# *Mentha piperita*: Essential Oil and Extracts, Their Biological Activities, and Perspectives on the Development of New Medicinal and Cosmetic Products

**DOI:** 10.3390/molecules28217444

**Published:** 2023-11-06

**Authors:** Nataliia Hudz, Lesya Kobylinska, Katarzyna Pokajewicz, Vladimira Horčinová Sedláčková, Roman Fedin, Mariia Voloshyn, Iryna Myskiv, Ján Brindza, Piotr Paweł Wieczorek, Jacek Lipok

**Affiliations:** 1Department of Pharmacy and Ecological Chemistry, University of Opole, 45-052 Opole, Poland; jacek.lipok@uni.opole.pl; 2Department of Drug Technology and Biopharmacy, Danylo Halytsky Lviv National Medical University, 79010 Lviv, Ukraine; 3Department of Biochemistry, Danylo Halytsky Lviv National Medical University, 79010 Lviv, Ukraine; kobylinska_lesya@meduniv.lviv.ua; 4Department of Analytical Chemistry, University of Opole, 45-052 Opole, Poland; katarzyna.pokajewicz@uni.opole.pl (K.P.); pwiecz@uni.opole.pl (P.P.W.); 5Faculty of Agrobiology and Food Resources, Slovak University of Agriculture in Nitra, 94976 Nitra, Slovakia; vladimira.sedlackova@uniag.sk (V.H.S.); jan.brindza@uniag.sk (J.B.); 6Department of Pharmacy and Biology, Stepan Gzhytskyi National University of Veterinary Medicine and Biotechnologies of Lviv, 79010 Lviv, Ukraine; roman_fedin@ukr.net; 7Department of Foreign Languages, Lviv Polytechnic National University, 79000 Lviv, Ukraine; marvollpi@gmail.com (M.V.); myskiviryna@i.ua (I.M.)

**Keywords:** *M. piperita*, essential oil, extracts, antimicrobial activity, antioxidant potential, anti-inflammatory activity, wound-healing effects

## Abstract

This review aims to analyze *Mentha piperita* L. as a potential raw material for the development of new health-promoting products (nutraceuticals, cosmetics, and pharmaceutical products). A lot of scientific publications were retrieved from the Scopus, PubMed, and Google Scholar databases which enable the study and generalization of the extraction procedures, key biologically active compounds of essential oil and extracts, biological properties, and therapeutic potential of *M. piperita*, along with perspectives on the development of its dosage forms, including combinations of synthetic active substances and herbal preparations of *M. piperita*. The results of this review indicate that *M. piperita* is a source rich in phytoconstituents of different chemical nature and can be regarded as a source of active substances to enhance health and to develop medicinal products for complementary therapy of various conditions, especially those related with oxidant stress, inflammation, and moderate infections. Essential oil has a broad spectrum of activities. Depending on the test and concentration, this essential oil has both anti- and prooxidant properties. Gram-positive bacteria are more sensitive to the essential oil of *M. piperita* than Gram-negative ones. This review also considered some facets of the standardization of essential oil and extracts of *M. piperita*. Among the identified phenolics of extracts were caffeic acid, rosmarinic acid, eriocitrin, luteolin derivates (luteolin-7-*O*-rutinoside, luteolin-7-*O*-glucoronide), and hesperidin. The concentration of these phenolics depends on the solvent used. This review also considered the relationships between the chemical component and biological activity. The results showed that the essential oil and extracts reduced inflammation in vitro by inhibiting the production of pro-inflammatory cytokines, such as tumor necrosis factor-alpha (TNF-α) and interleukin-6 (IL-6), and in vivo by reducing the paw edema induced using carrageenan injection in rats. Therefore, herbal preparations of *M. piperita* are promising medicinal and cosmetic preparations for their usage in skincare and oral cavity care products with antimicrobial, anti-inflammatory, and wound-healing properties. This plant can also be regarded as a platform for the development of antibacterial preparations and combined anti-inflammatory and cardioprotective medicinal products (synthetic active substances plus herbal preparations). This review could be considered for the justification of the composition of some medicinal products during their pharmaceutical development for writing a registration dossier in the format of Common Technical Document.

## 1. Introduction

Herbal and beekeeping products arouse a lot of interest from food scientists, pharmacists, chemists, food and herbal preparations producers, doctors, etc. [1,2,3,4,5,6,7,8,9,10,11,12,13,14,15,16,17,18,19].

*Mentha* species are famous as therapeutic herbs and have long served as natural herbal formulations. *Mentha piperita* L. (peppermint) is regarded as one of the best potential sources of biologically active substances for the food, cosmetics, and pharmaceutical industries [8,16,17,19,20,21]. *M. piperita*, a hybrid mint, is a cross-species between watermint (*Mentha aquatica* L.) and spearmint (*Mentha spicata* L.) [12,19,20,22]. Other *Mentha* species are also cultivated around the world for essential oil production [12,16,23,24].

The commercial growth and study of essential oil-bearing plants as sources of natural antioxidants and antimicrobial substances for food preservation and producing food and medicinal products for enhancing health are topical issues for agriculture, food chemistry, pharmaceuticals, and medicine. Such interest came from the studies related to the lower risk of developing cancer and cardiovascular diseases (CVDs) after the consumption of plant food, and to the microorganisms’ resistance to antimicrobial drugs [1,15,21,25,26].

*M. piperita* is a herbaceous aromatic and perennial herb native to Europe but cultivated in the northern USA, Canada, Asia, North Africa, and many other parts of the world [12,20,22,27]. It is very important from an economic perspective [8,9,12,18,27,28,29]. Moreover, *M. piperita* constitutes one of the most widely consumed single-ingredient herbal teas [15,30]. Its extracts and essential oil have different biological activities (antioxidant, antimicrobial, biopesticidal, anticancer, antiviral, antiallergic, anti-inflammatory, antihypertensive, urease inhibitory activity, etc.) [12,15,16,19,25,31,32]. *M. piperita* contains steroids, flavonoids and their glycosides, terpenoids, and phenolic acids [21,31,33,34,35,36]. Terpenoids and phenolics, including flavonoids, tannins, and phenolic acids, are believed to be responsible for the traditional pharmacological attributes of *M. piperita* [31]. Thus, *M. piperita* is a promising candidate for the development of new food, cosmetic, and pharmaceutical products. Its essential oil is the most popular natural product and one of the most produced and marketed essential oils worldwide. It is estimated that approximately 55% of essential oil produced is used for chewing gum production and a further 34% is utilized in toothpastes and other oral care product manufacturing [37,38,39]. 

The particular scent and flavor of plants of the genus *Mentha* were used for the purpose of masking the unpleasant taste of medicinal products long before the advent of pharmaceutical technology. The longevity and diversity of activities of these plants are a testament to their importance, still receiving a noble place in herbal medicine [29,40].

This review covers the detailed chemical profile of *M. piperita* essential oil and extracts. In addition, some biological activities of *M. piperita* are discussed in this review, especially antioxidant, anti-inflammatory, antimicrobial, anticancer, etc. Moreover, the review discusses some facets of the standardization of essential oil and extracts of *M. piperita* and the relationship between quantitative chemical components and antioxidant activity. Finally, this review is directed at the development of new pharmaceutical products with the basis of herbal preparations of *M. piperita* and could be considered for the justification of the composition of some medicinal products for writing registration dossier in the format of Common Technical Document.

## 2. Historical, Botanical, and Taxonomic Characteristics of *Mentha piperita*

The genus *Mentha* is a common taxonomic group in the Mediterranean flora [16,19,24,31,41,42]. *M. piperita* has been one of the most broadly used aromatic herbs for medicinal purposes since its discovery [16,23,27,31,37,40,43,44,45]. The aromatic qualities of *M. piperita* are connected with its hospitality in Greek mythology. *M. piperita* was used as a room deodorizer for spreading over floors with the purpose of reducing the unpleasant smell of hard-packed soils [23]. Pedanius Dioscorides (c. 40–90), a famous Greek physician and herbalist, first tried to determine the botanical classification of the plants in the genus *Mentha*. Several centuries later, Carl Nilsson Linnaeus (1707–1778), in his work *Species Plantarum* (1753), proposed a more formal classification of this genus [37,40]. This genus was also described and named by Jussieu in 1789 [24]. 

The genus *Mentha* was in a state of continuous change concerning its taxonomy, which is highly complex, and there is not always a consensus [44]. This genus was redefined to comprise 18–20 worldwide species and 11 hybrids, which are divided into four sections, on the basis of the chromosome numbers, phylogenetic analysis, and major components of essential oil [24,31,44,46]. Now the genus *Mentha* can be classified into 42 species, 15 hybrids, and hundreds of subspecies, varieties, and cultivars, and it is often divided into five sections: Audibertia, Eriodontes, Mentha, Preslia, and Pulegium [47,48]. The facets of the systematization of the genus *Mentha* were not the subject of this review. 

*M. piperita* has widespread overground and underground stolons. It has a square, erect, reddish, and branched stem that is relatively smooth or fringed, with a few spreading hairs, and is 2 or 3 feet in height. Its dark green leaves are arranged in opposite pairs, from oblong to lanceolate, approaching downy, and with sharp edges. The leaves are smooth or hairy on the underside, serrated, and borne on ciliated petioles. The flowers have prominent lip-like lower petals and are formed in false whorls, also known as verticillasters, with a purple color. Each flower has a two-lipped corolla with four lobes and its fruits have 1–4 seeds, covered with a stony layer. The whorls are few and lax, with the uppermost in a short, oblong, obtuse, reddish spike; and the lowermost remote, with the cymes shortly stalked. The bracts are subulate, the outer ones as long as the calyx. The pedicels are quite smooth. The calyxes are five-toothed, subulate, and erect. The corollas are four-cleft and tubular, with the broadest segment emarginated [20,23,27,44,49].

## 3. Chemical Composition of Essential Oil

The whole dried leaves of *M. piperita* contain not less than 12 mL/kg of essential oil. The cut dried leaves contain not less than 9 mL/kg of essential oil [33]. 

Essential oils are secondary metabolites of aromatic plants with a complex composition consisting of a mixture of different chemical compounds, mainly oxygenated compounds and hydrocarbons, which are accumulated in the oil ducts, resin ducts, glands, or trichomes of plants [27,42,46,50]. These compounds exert principal roles in the protection of plants against different assailants (insects, herbivores, and microorganisms). Moreover, these compounds have important pharmacological properties [16,50,51]. 

The essential oil of *M. piperita* is typically obtained through the steam distillation of aerial parts of the flowering plant [16,27,28,42,46,51]. The raw essential oil of *M. piperita* is usually subjected to further redistillation or rectification to remove unpleasant-smelling sulphur compounds, primarily dimethyl sulfide [39,52]. This essential oil is a colorless to light yellow or greenish oily liquid with a characteristic fresh odor and taste [51,53]. The principal components of the essential oil are *α*-pinene, *β*-pinene, sabinene, myrcene, *cis*-ocimene, *β*-caryophyllene, γ-terpinene, germacrene D, carvone (3.5%), *L*–menthol (38–70%), *L*–menthone (0.4–35%); isomenthone (1.5–10%), menthyl acetate (0–20%), eucalyptol (1,8-cineole) (0.4–12%), menthofuran (0.1–21%); limonene (0.6–4.5%), pulegone (traces–7%), and neoisomenthol [9,16,19,27,33,51,52].

The essential oil yield is one of the most important quality parameters [10]. As a rule, the essential oil yield ranges from 0.1 to 1% for air-dried parts [8]. However, this range can be widened to 0.5–4% [1,10,27]. In general, the yield and oil composition of *M. piperita* depends on many factors, among which are the plant genotype; plant origin; soil and climatic conditions; harvest stage and techniques used; isolation methods; weed protection; pedoclimatic conditions; stress conditions; ontogenetic stage of the plant; photoperiod; organic farming conditions, including use of plant growth regulators and fertilizers; and analytical procedure used; etc. [1,10,16,19,27,46,51,54,55,56,57,58,59]. For example, the yield of essential oil of *M. piperita* of Egyptian origin was (0.40 ± 0.01) mL or (1.21 ± 0.08) mL of essential oil per 100 g of fresh weight and air-dried herb, respectively [8]. The yield of essential oil of *M. piperita* cultivated in Santa Catarina (Brazil) was 2.5 mL per 100 g of air-dried herb (2.5% *v/m*) [1].

In general, the quality of essential oil of *M. piperita* mostly relies on the proportionate balance of its various components, especially menthol, menthone, menthofuran, and pulegone, and the essential oil’s quality is improved with higher menthol and menthone content and lower menthofuran and pulegone content [55,58]. The essential oil of *M. piperita* is characterized by the following quality indexes: consistency in the range of 0.900 to 0.916 g/cm^3^, an optical versatility in the range of −10 to −30°, a refractive index in the range of 1.457–1.467, and an acidity number of not more than 1.4 mg KOH/g [27]. Menthol has colorless, needle-shaped or prismatic crystals. It also has a strong odor and aromatic taste, followed by a sensation of cold when the air is drawn into the mouth. Menthol is supposed to be the component imparting a specific taste and odor to essential oil and the whole herb of *M. piperita* [27].

According to the E/S/C/O/P Monographs *Menthae piperitae* folium, the principal component is usually menthol in the form of (−)-menthol (30.0–55.0%), with smaller amounts of stereoisomers such as (+)-neomenthol (2.5–3.5%) and (+)-iso-menthol (approx. 3%) [33].

We summarized some publications from the point of view of the chemical composition and origin of the essential oil of *M. piperita* [19,27,59,60,61,62,63,64,65,66,67,68,69,70,71]. Sústriková and Šalamon, using gas chromatography–mass spectrometry (GC-MS) and GC-FID, determined the chemical composition of the essential oil of six samples of *M. piperita* grown in different parts of Eastern Slovakia (1999–2001). The following contents of the constituents were revealed: menthol (38.3–69.1%), menthone (0.4–20.9%), menthyl acetate (3.5–4.5%), iso-menthone (0.8–8.8%), linalool (0.6–5.1%), and limonene (2.50–6.70%). The essential oil of several cultivars of *M. piperita* had a very similar composition [27]. The menthol content varied from 32.92% to 39.65%, 34.29% to 42.83%, and 22.56% to 32.77% during the crop growth in three cultivars, viz., Kukrail’, ‘CIM-Madhurus’, and ‘CIM-Indus’, of *M. piperita* from Uttarakhand hills (India), which were analyzed using GC-FID [59]. Kehili et al. (2020) determined the content of menthol (53.29%) and found a high content of oxygenated monoterpene compounds (92.75%) in the essential oil of *M. piperita* grown in the Cherchell region (Tipaza, Algeria) [60]. Twenty-two components, accounting for 95.6% of the total essential oil, were identified in the *M. piperita* essential oil of Turkish origin using GC-MS. (+)-Menthol (38.1%), menthol (35.6%), neomenthol (6.7%), and eucalyptol (1,8-cineole) (3.6%) were the main components [61]. Al-Mijali et al. (2022) determined the main compounds of the essential oil of Moroccan *M. piperita* using GC-MS. Among them were menthone (29.24%), levomenthol (38.73%), menthol (2.71%), and eucalyptol (6.75%) [62]. The chemical composition of the essential oil from *M. piperita*, analyzed using GC/FID and GC-MS, showed that the main constituents were menthol (40.7%) and menthone (23.4%) [63]. Using GC-MS, Gharib and Silva determined such components as *p*-menthone (36.6%) and neo-menthol (40.5%) [8]. It is worth mentioning that there are some problems with the identifications and nomenclature of the isomers of menthol in some publications related to *M. piperita* as, for instance, (+)-menthol stated in paper [61] is not specific for *M. piperita*.

The essential oil composition differs depending on the ontogenetic stage of the plant and growing conditions. For instance, young plant oils generally contain a higher concentration of menthone (40–55%) than menthol (20–30%). Later, menthone is converted to menthol, and during the blooming period the content of the latter considerably increases. After the blooming period, in October or November, the menthone content considerably decreases. The toxic components, pulegone and menthofuran, are the most abundant in essential oil during the flowering stage, with menthofuran present mostly in flowers (even up to a content of 15%) [52,64,65]. In other words, *M. piperita* harvested at the flowering phase has a higher content of menthol compared to during the bud phase, when larger amounts of menthone are produced. In addition, stress conditions connected to light, temperature, and moisture status tend to promote the accumulation of pulegone and menthofuran. Moreover, menthofuran synthase expression (particularly overexpression), which leads to the production of high levels of menthofuran, results in a notable decrease in pulegone reductase message, accompanied by a decrease in reductase activity, with the consequence of increasing in the pulegone concentration in the essential oil [58].

Using GC-MS, Nilo et al. detected 2.3% of menthyl acetate, 3.9% of eucalyptol (1,8-cineole), 10.5% of pulegone, 12.0% of menthofuran, 12.0% of isomenthol, 14.1% of menthol, and 31.4% of menthone. These authors considered menthofuran as an original marker to identify an authentic essential oil of *M. piperita* [1]. In addition to this statement, menthofuran was found at low levels (traces) in the essential oil of *M. arvensis* L. [1], and was even absent in *M. spicata* [9,16,61], *M. × gentilis* L. [9], *M. crispa* L. [9], and *M. gracilis* R.Br. [16]. Moreover, according to the requirements of the monograph of the European Pharmacopeia, using GC-FID analysis it should be present, but its content can range widely (1.0-8.0%) [53]. However, according to the analysis performed in some publications, it was revealed that it could be non-identified in the essential oil of *M. piperita* [8,19]. Finally, essential oil *M. spicata* can be considered adulterant for peppermint [28].

Menthol is a cyclic monoterpene alcohol with three asymmetric carbons [58,69,72]. The main stereoisomer of menthol found in nature and in *M. piperita* is levomenthol ((−)-menthol), with a configuration of (1R,3R,4S). Other isomers include isomenthol (0.3–0.4%), neomenthol (0.1–6.5%), and neoisomenthol (0.2–1.5%) [38,73,74]. Menthol isomers, especially levomenthol, have a widely appreciated strong, cooling, and refreshing aroma and flavor. Its cooling effect is attributed to its ability to activate the cold-sensitive transient receptor potential cation channel (TRPM8) [73,75]. (−)-Menthone, the second most abundant monoterpenoid in the essential oil of *M. piperita*, is a critical intermediate in menthol biosynthesis and possesses a minty aroma and various acknowledged biological activities. (−)-Menthone is directly synthesized from (+)-pulegone through the reduction of the C2–C8 alkene double bond, catalyzed using (+)-pulegone reductase [76]. (+)-Pulegone is the precursor of (−)-menthone, (−)-menthol, and (+)-menthofuran [58,76]. Pulegone and its metabolite menthofuran can adversely affect the quality of the essential oil [58,72,75,77]. These components are known to display hepatotoxicity and, therefore, their content is limited by some regulatory authorities. According to the EC1334/2008 regulations, a joint maximum daily intake of 0.1 mg/kg body weight was established for pulegone and menthofuran. The content of menthofuran in foods and beverages should not be more than 20 mg/kg, with the exceptions of mint/peppermint-flavored alcoholic beverages (100 mg/kg), mint/peppermint-flavored confectionery (200 mg/kg), and mint/peppermint-flavored chewing gum (1000 mg/kg). The content of pulegone in foods and beverages should not be more than 20 mg/kg, with the exceptions of mint/peppermint-flavored alcoholic beverages (100 mg/kg), mint/peppermint-flavored confectionery 100 mg/kg, intensely strong mint/peppermint-flavored confectionery (200 mg/kg), and mint/peppermint flavored chewing gum 30 mg/kg [77]. Therefore, it is preferred that the essential oil of *M. piperita* contain a high concentration of menthol, moderate levels of menthone, and low amounts of pulegone and menthofuran [58,65].

The essential oil of *M. piperita* is among the most important essential oils used in the pharmaceutical and cosmetics industries [16,28,56,78,79,80,81]. Therefore, there is an International Standard Organization (ISO) standard ISO 856:2006 “Oil of peppermint (*Mentha* × *piperita* L.)” [82]. This essential oil is also included in many pharmacopeias [37,39] and WHO monographs on selected medicinal plants [83]. The United States Pharmacopeia–National Formulary (USP-NF) [84] peppermint oil monograph does not set any specifications for the chromatographic profile of the essential oil of *M. piperita*. The normative ranges for the main components of this essential oil are presented in Table 1. The European Pharmacopeia regulates fewer components and generally permits wider ranges than the ISO [53].

It should be mentioned that the chemical composition of the essential oil of *M. piperita* and its yield depend on plant hormones, fertilizers, and abiotic and biotic factors. Among the abiotic factors are climate, soil, sunlight, salinity, etc. [43,45,46,56]. For instance, salinity stress reduced the plant stem and root length, the internode length, the shoot and root fresh and dry weight, and the yield of essential oil, while there was an increase in the proline content, possibly because of a decrease in proline oxidase activity in saline conditions. Four levels of salinity were formed by four solutions containing 0, 50, 100, and 200 mmol/L of sodium chloride for irrigation plantings in a greenhouse [43]. Among the consequences of soil moisture stress, at which the plants grew at 75% and 50% FC, there was a decrease in the fresh and dry weight of leaves, leaf width and length, and shoot length compared to the control plants; however, there was an increase in the TPC and TFC [45].

Plant hormones had an impact on the percentage of particular compounds in the composition of volatile organic substances. Some compounds or their combination were used for the studies (6-benzyloadeninopurine + indolyl-3-acetic acid (MS-1), 2-isopentinloadenine + indolyl-3-butyric acid (MS-2), indolyl-3-butyric acid (MS-3), 6-benzyloadeninopurine (MS-4), along with a control medium without the addition of plant hormones (MS-K). The menthofuranolactone share was significantly increased in sample MS-2 while the limonene and eucalyptol mixture share was decreased. It should be noted that the authors used the special chemotype of *M. piperita*, which was characterized by high amounts of limonene, eucalyptol (1,8-cineole), and menthofuranolactone. In their studies, there was no menthol or menthone. These authors explained the lack of most characteristic components of essential oil of *M. piperita* (menthone, menthol, and their derivatives) as a result of the very early stage of the plant development in the in vitro culture [56]. In general, this partly conforms to the statement that the menthol content increases with the leaf and oil gland’s maturity [58]. Menthol and menthone contents were different in response to the different fertilization treatments. The highest percentage of menthol was obtained in response to the treatment of the combination of organic fertilizers (poultry manure + cattle manure) and effective microorganisms (41.5%). Contrastingly, the lowest percentage of menthol was obtained in the plants fertilized with cattle and poultry manures as organic fertilizers (39.1% and 39.6%, respectively). However, this content was higher compared to the control (38.2%) [57].

The studies with plant growth regulators and fertilizers indicated that farming conditions should be standardized for growing *M. piperita* for the pharmaceutical and food industries.

## 4. Chemical Composition of Extracts of *Mentha piperita*

The hydroalcoholic and other extracts of *M. piperita* contain flavonoids from different groups (epicatechin; rutin; quercetin; naringenin; naringenin-7-*O*-glucoside; kaemferol; hesperidin (3′,5,7-trihydroxy-4′-methoxyflavanone 7-rutinoside); luteolin (3′,4′,5,7-tetrahydroxyflavone); luteolin-7-*O*-glucoside; eriocitrin ((*S*)-3′,4′,5,7-tetrahydroxyflavanone-7-[6-*O*-(α-L-rhamnopyranosyl)-β-d-glucopyranosid], apigenin, etc.) [35,42,81,85,86,87]; and phenolic acids, including cafeic acid [85,86,87], rosmarinic acid [17,86,87,88], trierpenoids (ursolic acid), etc. [85]. Their amounts in extracts depend first of all on the solvent chosen for extraction [86]. The chemical structure of main flavonoids of *M. piperira* is provided on Figure 1.

The TFC content was in the range of 29.2 to 53.2 mg of catechin equivalents in one gram of dry mass depending on conditions of the vacuum drying process of *M. piperita*. Therefore, processes of drying are important for the pharmaceutical industry, as they have an impact on the content of biologically active substances [21]. The reflux extraction of *M. piperita* with methanol in 45 min at a temperature of 70 °C resulted in an extraction yield of 0.9%. The obtained extract, evaluated using high-performance liquid chromatography (HPLC), indicated that rosmarinic acid was present at a high concentration (1.9 mg/mL in the extract) [17].

Eriocitrin (2.7–182.6 mg/g), luteolin-7-*O*-glucoside (3.2–90.8 mg/g), and rosmarinic acid (10.6–176.8 mg/g) were the most abundant components identified in the leaves of *M. piperita*, whilst naringenin-7-*O*-glucoside was the least abundant component found only in the ethylacetate extract (0.8 mg/g). Eriocitrin was found in the largest and smallest amounts in the acetonitrile and aqueous extracts, respectively (182.6 and 2.7 mg/g). Luteolin-7-*O*-glucoside was found in the largest and smallest amounts in the methanolic and aqueous extracts, respectively (90.8 and 3.2 mg/g). Among the identified compounds were caffeic acid (1.4–1.8 mg/g), hesperidin (3.8–32.6 mg/g), eriodictyol, and luteolin. Moreover, these studies showed that methanol and water were the solvents which extracted the most and the least active substances, respectively. In addition, the ethylacetate, acetonitrile, and aqueous extracts were the most active in such tests as the iron (III) reduction and DPPH test, ABTS test, and hydroxyl free radical scavenging properties. The aqueous and dichloromethane extracts had the most active antioxidant activity in the β-carotene-linoleic acid bleaching inhibition assay. The light petroleum, methanolic, and aqueous extracts showed moderate iron (II) chelating activity [86]. According to the requirements of the Pharmacopeias, it is necessary to avoid acetonitrile, dichlormethane, and methanol in the manufacture of medicinal products [53,89]. Therefore, these studies have more of a theoretical meaning concerning the dependence of the amounts of extractive substances on the nature of the solvents used.

In another study, it was reported that the aqueous extracts (teas) of *M. piperita* contain phenolic compounds such as hesperidin (18.64 ± 0.16 mg/100 mL of the tea), rosmarinic acid (4.91 ± 0.18 mg/100 mL of the tea), eriocitrin (1.55 ± 0.08 mg/100 mL of the tea), chlorogenic acid (traces), and luteolin 7-*O*-β-glucuronide 3.68 ± 0.01 mg/100 mL [1]. This study was in line with Dorman et al. [86]. Similar results were obtained for *M. piperita* of Greek origin. As a result of the maceration of *M. piperita* using a combined solvent (90% of water and 10 % of ethanol), the following phenolics were extracted: luteolin 27.64 ± 0.32 ppm, quercetin 6.25 ± 0.46 ppm, apigenin 0.10 ± 0.03 ppm, rutin 8.32 ± 0.59 ppm, eriodictyol 32.23 ± 0.75 ppm, rosmarinic acid 6.03 ± 0.52 ppm, *p*-coumaric acid 35.21 ± 1.02 ppm, ferulic acid 31.92 ± 0.83 ppm, caffeic acid 116.89 ± 0.28 ppm, hydroxybenzoic acid 30.86 ± 0.92 ppm, and benzoic acid 41.92 ± 0.61 ppm [87].

Athanasiadis et al. confirmed the presence of phenolics such as caffeic acid, eriocitrin, luteolin-7-*O*-rutinoside, luteolin-7-*O*-glucoronide, hesperidin, rosmarinic acid, and luteolin derivate. This fact has been observed in many scientific publications [87,90,91]. Sinapic acid, gallic acid, syringic acid, kaemferol, quercetin, epigallocatechin, and quercetin-rhamno-di-hexoside were found in *M. piperita* of Pakistani origin [91]. Athanasiadis et al. also revealed that adding *β*-cyclodextrin increased the yield of some polyphenols of *M. piperita* in aqueous extracts compared to aqueous extracts without *β*-cyclodextrin. Moreover, these authors compared the *β*-cyclodextrin aqueous extracts to the other aqueous and ethanolic extracts. It was confirmed that there were three principal components in all the extracts, namely, eriocitrin, luteolin-7-*O*-rutinoside, and rosmarinic acid. In general, ethanol extracted the phenolics the best. However, the *β*-cyclodextrin aqueous extract was richer in some phenolics compared to the aqueous extract [90]. Chlorogenic acid, rosmarinic acid, epicatechin, quercetin, gallic acid, epigallocatechin, syringic acid, kaempferol, and caffeic acid were identified in the methanolic extract and aqueous extracts of *M. piperita* [92].

Therefore, caffeic acid, rosmarinic acid, eriocitrin, luteolin-7-*O*-rutinoside, luteolin-7-*O*-glucoronide, and hesperidin can be used as the markers for the quantitative and qualitative analysis of extracts of *M. piperita* using the spectrophotometric method for measuring TPC and TFC, high-performance thin-layer chromatography, and HPLC as it was previously published for the extracts of *Satureja montana* L. [4]. Moreover, some authors attribute anti-inflammatory and antiviral activity of extracts of *M. piperita* to rosmarinic acid, luteolin-7-*O*-rutinoside, and hesperidin [12].

Among the main pharmacological properties of *M. piperita* extracts and essential oil are antioxidant [9,12,16], anti-inflammatory [19,93], immunomodulatory [85], hepatorenalprotective [94], and antimicrobial activities [7,12,20,95].

## 5. Antioxidant Activity

*M. piperita* extracts and essential oil possess significant antioxidant potential and related biological activities, among which is anti-inflammatory activity [9,12,16,17,19,60,78,86,94,96].

Numerous studies have demonstrated the ability of herbal preparations to scavenge free radicals, donate electrons or hydrogen, and inhibit lipid peroxidation in vitro and in vivo [16,78,86,94]. A disproportionate amount of ROSs and the absence of their scavenging systems in cells result in oxidative stress, increasing the risk of the development of various diseases such as diabetes, CVDs, cancer, neurodegenerative diseases, and liver cirrhosis, and speeding up the aging process [86,94]. Releasing free radicals initiates lipid peroxidation, leading to membrane destruction, and subsequently induces an inflammatory response by producing mediators and chemostatic factors [12,16,17,85]. It is supposed that herbal preparations will decrease cellular damage which appears as a result of excessive phagocyte activation and production of ROSs [12].

Among the tests used to measure the antioxidant activity of *M. piperita* is the DPPH test (free radical scavenging activity) [1,12,18,20,63,81,88], β-carotene-linoleic acid assay, NO radical scavenging, reducing power [20], assessment of linoleic acid peroxidation through measurement of thiobarbituric acid reactive substances [1], etc. Antioxidants scavenge radicals through different mechanisms—hydrogen atom transfer or single electron transfer mechanism; or a combination of both mechanisms [16,17,97,98]. The ABTS and ORAC tests are based on the hydrogen atom transfer mechanism. The DPPH and FRAP tests are based on the single electron transfer mechanism. However, it is very difficult to distinguish between the two mechanisms [17,96,97,98]. In addition, the mechanism of the DPPH test can be based on the reaction of DPPH with ROSs and unsaturated hydrocarbons, joining hydrogen atoms from C–H bonds, for instance, from the C–H bonds of α-pinene [1,8,96]. DPPH test is sensitive and is widely used in food chemistry [1,2,12,17,19,31,86,96,97,98].

The antioxidant activity of the ethanolic extract of *M. piperita* has the capacity to reduce cellular damage. Gharib and Silvia compared the total phenolic content (TPC) and radical scavenging activity in the DPPH test of four essential oils obtained from the fresh herbs of *Majorana hortensis* Moench, *M. spicata* L., *M. piperita* L., and *Rosmarinus officinalis* L. [8]. It was established that IC_50_ ranged from 59.2 to 65.4 µg/mL and the best radical scavenging activity was measured for *M. piperita* (59.2 µg/mL) and *R. officinalis* (62.5 µg/mL). The radical scavenging activity of the four essential oils was much lower compared to tertiary butyl hydroquinone (TBHQ) (IC_50_ = 29.8 µg/mL). The TPC was in the range of 0.042 to 0.163 mg/100 µL for the essential oils of the above-mentioned species. The best radical scavenging activity of the essential oil of *M. piperita* could be explained by the highest TPC content (0.163 mg of gallic acid equivalents per 100 µL of the essential oil). The authors attributed the high radical scavenging activity of the essential oils of *M. piperita* and *R. officinalis* to monoterpenes such as α-pinene, 1,8-cineole, camphor, neo-menthol, and borneol [8].

However, Nilo et al. consider that DPPH discoloration is unable to completely reflect the antioxidant activity of the essential oil of *M. piperita.* DPPH reduction can be related to the abstraction of a hydrogen atom from α-pinene and the formation of prooxidant species. Secondly, these authors specified that the essential oils of some species in the genus *Mentha* can show prooxidant or antioxidant activity depending on the oil composition and the test used to measure oxidant activity. In the forced oxidation test with soybean oil, the tested mint essential oils (*M. piperita* and *M. arvensis*) showed prooxidant activity in contrast to the modest antioxidant activity in the DPPH test. The TPC (mg/g) of the essential oils from *M. piperita* and *M. arvensis* was 0.34 ± 0.18 and less than 0.10, respectively. The IC_50_ values were 13.58 ± 0.16 mg/mL and 26.72 ± 0.16 mg/mL for *M. piperita* and *M. arvensis*, respectively. Therefore, the infusions from dried leaves of *M. piperita* and *M. arvensis* showed lower IC_50_ in the DPPH test and lower TPC than the commercial sachets on the base of *M. piperita*. This issue is very controversial and it can be explained by the possible addition of a herbal material from other species of the genus *Mentha* to the commercial sachets. Moreover, Nilo et al. reported some discordances in their studies. They noticed the low menthofuran content and low yield of essential oil (0.5–0.7%) compared to the values of 2.5% and 4% reported for the oil yield of *M. piperita* and *M. arvensis*, respectively [1]. Furthermore, such studies should have covered an anatomical examination of the content of the commercial sachets.

The essential oils of five *Mentha* species (*M. spicata*, *M.*
*×*
*gentilis* L., *M. crispa* L., *M. piperita*, and *M.*
*×*
*piperita* L.) were studied for their antioxidant capacity using the test of differential pulse voltammetry. This test works on the principle of the reduction in the limiting current value of the oxygen electroreduction. The charge transfer processes go through some stages with the formation of the superoxide anion-radical of oxygen in the absence of *Mentha* extracts. The species *M.*
*×*
*gentilis* L. had the highest antioxidant capacity value (the rate constant was 270.1 ± 0.9 mL/g), *M.*
*×*
*piperita* had the second highest value (the rate constant was 155.9 ± 0.7 mL/g) and *M. piperita* had the third highest value (the rate constant was 122.4 ± 0.2 mL/g) [9]. However, there was no clear reason why the authors described the same species grown in different parts of Brazil in two different ways (*M. piperita* and *M.*
*×*
*piperita*).

The Turkish scientists evaluated three mint genotypes (*M. spicata* L., *M. canadensis* L., and *M. piperita*) using methanol/chloroform (3:1) extracts and achieved the following results. The TPC ranged from 9.39 ± 1.42 to 28.27 ± 3.95 μg of gallic acid equivalents per gram of dry weight and FRAP activity ranged from 280.73 ± 58.76 to 577.09 ± 46.02 μmol trolox equivalents per gram of dry weight for *M. spicata*, while *M. piperita* had higher TEAC activity (771.58 ± 3.22 and 800.02 ± 1.10 μmol trolox equivalents per one gram of dry weight). The FRAP and TEAC activities of the selected clones were correlated with TPC (r = 0.77; 0.73 respectively) [98].

According to Wu et al., testing cellular-based antioxidant activity is biologically more important than performing chemical-based antioxidant assays. Dichlorofluorescin in cultured cells is oxidized to fluorescent dichlorofluorescein in the presence of intracellular peroxyl radicals. The essential oils of *M. piperita*, *M. spicata*, and *M. gracilis* eliminated experimentally induced intracellular peroxyl radicals and inhibited the formation of fluorescent dichlorofluorescein. The supplementation of these three essential oils in the culture media significantly decreased the area under the curve compared to non-supplemented control samples, suggesting the protective activity of the essential oils against cellular oxidative damage. The fact of compromised cellular antioxidant effects with a higher concentration of the essential oils further supported the prooxidant activity of mint essential oils [16]. Wu et al. explained that the prooxidant activity of the mint essential oils was due to α-pipene, linalool, and λ-terpinene, which can be subjects of autoxidation. As a result, they produce reactive alkyl radical through prooxidant activities [16].

The effectiveness of three essential oils of *Mentha* species to alleviate acute systemic oxidative damage was evaluated in vivo using *Caenorhabditis elegans*. *M. piperita*, *M. spicata*, and *M. gracilis* were collected in the Midwest region of the USA before the full flowering stage. These essential oils showed high DPPH scavenging activity in a TEAC assay and Fe^3+^ reducing activity. In comparison with *M. spicata* and *M. gracilis* essential oils, *M. piperita* essential oil had the lowest half maximal effective concentration (EC_50_) in DPPH and TEAC assays, and a reducing power assay. All the essential oils similarly mitigated the chemical-induced lipid peroxidation in the liver tissue in a dose-dependent manner. The highest cellular antioxidant activity was observed at a dose of 5 µg/mL for the *M. piperita* essential oil and at 100 µg/mL for the *M. spicata* and *M. gracilis* essential oil. In these studies, the authors observed compromised cellular activities at higher concentrations of the essential oils, namely, a prooxidant effect. Increasing the concentrations of the essential oil of *M. piperita* from 5 to 200 µg/mL was accompanied by enhanced prooxidant activity. In addition, these authors revealed cytotoxic activities of the mint essential oils concerning porcine epithelial cells (IPEC-J2). The addition of 25 µg/mL of *M. spicata* and *M. gracilis* essential oils increased the cellular concentrations of glutathione in H_2_O_2_-treated IPEC-J2 cells. The supplementation of 100 µg/mL of *M. piperita* or *M. gracilis* essential oils significantly increased the survival rate of *C. elegans* in response to H_2_O_2_-induced oxidative stress. The protective effect was comparable to that of the supplementation of 10 µg/mL of ascorbic acid. However, *M. spicata* essential oil did not reduce the death rate within the same supplementation dose (10–200 µg/mL) [16]. These and other studies support the idea that in developing antimicrobial and anti-inflammatory herbal medicinal products it is necessary to take into account the prooxidant and oxidant properties of the essential oil of *M. piperita* and their correlation with used concentrations.

In another study, the chloroform extract and essential oil of *M. piperita* showed almost similar antioxidant potencies (about 90%) in a test with peroxidase and 2,2-azino-bis-3-ethylbenzthiazoline-6-sulfonic acid. Very similar trends were observed in the DPPH test and reducing power absorbance test of both essential oil and chloroform extract [20]. It is worth mentioning that values of antioxidant activity of approximately 90% need the obligatory modification of an analytical procedure (dilution of samples) because there is no change or difference in antioxidant activity if the concentration of samples is too high [2]. The aqueous extract showed the least activity in the three tests: the antioxidant activity was 734 nm, the DPPH scavenging activity was 70.3% ± 6.1%, and the reducing power test found an absorbance of 0.4 ± 0.3. The rest of the leaf extracts showed an antioxidant capacity, DPPH scavenging activity, and the reducing power exhibited intermediate values between those of chloroform and the aqueous extracts. The IC_50_ of the *M. piperita* essential oil, determined using the DPPH scavenging method, was found to be 15.2 ± 0.9 µg/mL compared to the positive control BHT, which was 6.1 ± 0.3 µg/mL [20]. These studies indicated that ethanolic extracts of *M. piperita* are richer in biologically active substances compared to aqueous ones, which conforms with the findings of the studies performed by Athanasiadis et al. and Dorman et al. [86,90].

Therefore, ethanolic extracts can be components used in different dosages or cosmetic forms for the treatment of conditions or care of the oral cavity, including halitosis. Moreover, considering that ethylacetate belongs to toxicity class 3 [53], it can be safely used in the extraction of biologically active substances of *M. piperita* and its obtained extracts following evaporation (dried extracts) can be a component of tablets, capsules, sprays, gels, toothpastes, etc., with antimicrobial and/or deodorizing activity. Considering the relatively high content of TPC and TFC, and the antioxidant activity and antimicrobial activity of aqueous extracts of *M. piperita* [20,86], the issue of the usage of aqueous dried extracts is topical as well. Aqueous dried extracts can also be used in different dosages and cosmetic forms for the complementary treatment of some conditions or care of the oral cavity, including halitosis.

Mallick et al. studied *M. piperita* for future exploitation through in vitro propagation to evaluate the antioxidative activity of the tissue culture-derived clones and compared it with that of in vivo field-grown plants. It was observed that the antioxidant activity of the field-grown plants was well-maintained in the tissue culture-derived plant. The TPC was 6.24 ± 0.03 mg of gallic acid equivalents per gram, the content of rosmarinic acid was 7.28 ± 0.28 mg/g, and the TFC was 49.29 ± 0.48 mg of quercetin equivalents per gram. Meanwhile, for the extract of the field-grown plants, the TPC was 5.82 ± 0.03 mg gallic acid equivalents per gram, the content of rosmarinic acid was 7.0 ± 0.08 mg/g, and the TFC was 48.54 ± 0.24 mg quercetin equivalents [88]. Therefore, tissue culture-derived clones can be considered as an alternative to naturally grown threatened plants.

The essential oil of *M. piperita* showed a stronger antioxidant impact on the OH• radical compared to DPPH (the IC_50_ was 0.26 and 860 µg/mL, respectively). The scavenging of the radicals OH• is connected to the hydrogen-donating ability of menthol. This essential oil also had antioxidant activity in a model linoleic acid emulsion system, inhibiting conjugated dienes and linoleic acid secondary oxidized product generation by 52.4% and 76.9%, respectively (at a concentration of 0.1%). Conjugated dienes are formed in the early stages of the process of linoleic acid autooxidation [63]. The issue of anti- and prooxidant activity in the test of forming conjugated dienes is disputable as the tested essential oils of *M. piperita* and *M. arvensis* showed prooxidant activity in the forced oxidation test with soybean oil [1].

Nickavar et al. evaluated the antioxidant properties using DPPH and ABTS tests and measuring the TPC of the ethanolic extracts from five *Mentha* species *(M. longifolia* (L.) Huds., *M. piperita* L., *M. pulegium* L., *M. rotundifolia* (L.) Huds., and *M. spicata* L.). *M. piperita* exhibited the strongest activity as a DPPH• scavenger (IC_50_ = 13.32 (12.12–14.64) μg/mL), while *M. spicata* had the weakest activity (IC_50_ = 87.89 (81.66–94.59) μg/mL. On the other hand, all the extracts were active in the ABTS assay and no significant differences were observed in this assay. *M. piperita* showed the highest TPC (433.60 ± 19.62 μg/mg), whereas *M. spicata* had the lowest content (150.9 ± 5.14 μg/mg). A high correlation was found between the DPPH• scavenging activity and the TPC of the extracts (r^2^ > 0.989) [99].

*M. piperita* leaf essential oil protects the liver and kidney from CCl_4_-induced oxidative stress and thus substantiates the beneficial effects traditionally attributed to this plant [79,94]. The in vitro antioxidant activity of the essential oil was lower than that of silymarin. The pretreatment with the essential oil at doses of 15 and 40 mg/kg prior to CCl_4_ significantly reduced the stress parameters (serum alanine aminotransferase (ALT), aspartate aminotransferase (AST), alkaline phosphatase (ALP), lactate dehydrogenase (LDH), γ-glutamyl transpeptidase (γGT), urea, and creatinine) compared to the CCl_4_-only group. However, the pretreatment in animals with the essential oil at a dose of 5 mg/kg did not have a significant effect on ALT, AST, ALP, LDH, γGT, urea, or creatinine levels in a CCl_4_-induced stress. A significant reduction in hepatic and kidney lipid peroxidation and an increase in antioxidant enzymes (liver and kidney superoxide dismutase (SOD), catalase (CAT), and glutathione peroxidase (GPx)) was observed after the pretreatment with the essential oil (at 40 mg/kg) compared to CCl_4_-treated rats. Furthermore, the pretreatment with the essential oil at 40 mg/kg could also markedly improve the histopathological hepatic and kidney lesions induced through the administration of CCl_4_ [94].

In another study, purified rosmarinic acid from methanolic extract of *M. piperita* had a high antioxidant potential, which was greater than 95% for a DPPH assay and 87.83% for a H_2_O_2_ scavenging assay (at a concentration of 100 μg/mL). Rosmarinic acid can compete with unsaturated acids for binding to lipid peroxyl groups, stopping the chain process of lipid peroxidation. This study could be considered as at least a partial explanation for why extracts of *M. piperita* demonstrate antioxidant potential in different tests related to the neutralization of free radicals. In addition, this possible instructional technique proved to be a quick and successful method for measuring the antioxidant properties of rosmarinic acid [17].

All of the analyzed publications provide essential oil and extracts of *M. piperita* as DPPH•-, ABTS•+, and OH•-scavengers. During in vivo tests there are pro- and antioxidant effects of essential oil depending on the concentration used. This topic needs further study.

The generalized information about the results of antioxidant activity of essential oil and extracts of *M. piperita* provided by different authors is presented in Table 2.

As can be seen from Table 2, it is impossible to compare the values provided in different papers because of the different analytical procedures used by different authors, including various reference substances used for positive controls. It is only possible to compare the results provided in the same paper for different plants, extracts, tests, or extracts and essential oils. In addition, the data in this table show that anti- or prooxidant activity depends on the concentration of the essential oil.

## 6. Anti-Inflammatory Activity

Studies of plant-based extracts and essential oils as anti-inflammatory preparations arouse a lot of interest, considering the adverse reactions of non-steroidal anti-inflammatory medicinal products [12,16,19,60,78,80,100,101,102,103,104,105,106].

Inflammation is a natural complex biological response to pathogens, damaged cells (injury), or irritants, which involves immune cells, blood vessels, excessive phagocyte activation, molecular mediators, and the production of hydroxyl, superoxide anions, and non-free-radical species such as H_2_O_2_ [12,78,106]. In addition, inflammation is usually connected to edema and pain at the place of an injury or wound [60,101]. Moreover, unrestricted inflammation and chronic inflammation can lead to various diseases, such as rheumatoid arthritis, asthma, cancer, and CVDs, or even to the loss of functions or tissue structure [10,12,19,60,106]. Macrophages regulate inflammation and the immune response by releasing proinflammatory cytokines (TNF-a, IL-1b, and IL-6), nitric oxide (NO), and prostaglandin E2 (PGE2) [12].

Non-steroidal anti-inflammatory medicinal products reduce inflammatory effects, inhibiting COX which converts arachidonic acid into the prostaglandins and thromboxanes involved in inflammation, pain, and platelet aggregation [105]. However, their administration is associated with a high risk of adverse reactions, especially upper gastrointestinal tract lesions, cardiovascular and kidney toxicities, gastritis, oesophagitis, peptic ulcers, and their severe complications including bleeding and perforation [60,100,101,105]. For these reasons, it is very topical to manufacture safer combined medicinal products on the basis of synthetic active pharmaceutical ingredients and herbal preparations with fewer side effects, which combine antioxidant and anti-inflammatory activities.

*M. piperita* extracts and essential oil exhibit anti-inflammatory activity, which can help to reduce inflammation and prevent the development of chronic diseases [12,19,60,102].

It was revealed that treatment of the murine macrophage cell line RAW 264.7 with LPS alone substantially enhanced the secretion of proinflammatory mediators and cytokines including NO, TNF-α, IL-6, and PGE2. The ethanolic extract of *M. piperita* significantly reduced the LPS-induced NO production in a concentration-dependent manner and suppressed the secretion of the proinflammatory cytokines (TNF-α, IL-6, PGE2) in the cells stimulated with LPS compared to the cells stimulated without adding the extract. The TPC and TFC were 325.84 ± 14.17 µg of gallic equivalents and 118.92 ± 17.09 µg of rutin equivalents per mg of the extract, respectively. Such biological activities could be attributed to the synergistic action of the constituents of *M. piperita*, including substances of the phenolic structure [12]. Similar results were obtained with the essential oil of *M. piperita* of Chinese origin in a croton oil-induced mouse ear edema model. This essential oil reduced the LPS-induced NO production [19].

A cream with the essential oil of *M. piperita* added (0.5% *w*/*w*) was tested using the circular excision wound model and histological examination. The topical application of this cream induced a significant decrease in the unhealed wound area rate between the sixth and ninth days of treatment in comparison with the vehicle and madecassol at the sixth day. Moreover, a better epithelization, fibroblast population, and collagen deposition were observed in the animals treated with the cream in comparison with the vehicle and madecassol. Furthermore, in a carrageen-induced paw edema test the anti-inflammatory activity was revealed for the essential oil administered orally at doses of 20 and 200 µL/kg. The wound-healing and anti-edematogenic activities were attributed to menthol [60]. These findings support the anti-inflammatory and wound-healing properties of the essential oil of *M. piperita*. In addition, similar activity was observed from the oral administration of essential oil of *M. piperita*, which reduced cellular infiltration and subcutaneous xylene-induced ear edema [60]. A similar anti-inflammatory effect in the carrageen test was also observed for *M. spicata* essential oil [101].

The essential oil of *M. piperita* chocolate showed higher activity than *M. piperita* with regard to scavenging NO radical activity, with IC_50_ values of 0.31 and 0.42 μL/mL, respectively. *M. piperita* chocolate also had higher anti-inflammatory activity than the *M. piperita* oil, determined using a 5-lipoxygenase inhibition assay. In this test, linoleic acid is enzymatically converted to a conjugated diene by 5-lipoxygenase, which results in a continuous increase in absorbance at 234 nm [80].

Immunomodulatory effects of herbal preparations to a certain extent are related to their antioxidant and anti-inflammatory activities. The *M. piperita* essential oils from two cultivars (RAC 541 and Laimburg) grown in two locations of northern Italy were investigated for their effects on human polymorphonuclear leukocytes (PMNs) and peripheral blood mononuclear cells (PBMCs), and for their antioxidant activity and chemical composition. PMNs play a principal role in the first line of defense against the invasion of microorganisms. However, they also contribute to organ damage induced through excessive acute inflammatory responses. All the essential oils contained menthol and menthofuran in the range of 45% to 52% and 0.06% to 4.25%, respectively. The essential oils showed antioxidant activities at concentrations of ≥0.01 µg/mL [102]. However, it is not clear why the authors showed on the graphs the negative values of antioxidant activity in the DPPH test. As a rule, such values have compounds with prooxidant activity [97]. All the essential oils were cytotoxic for PMNs and PBMCs only at concentrations greater than 0.1 µg/mL in the MTT test. In the range of 0.01–0.0001 µg/mL, the essential oils did not affect the spontaneous oxidative bursting of PMNs. However, there are some examples of essential oils significantly increasing ROS production and proliferation in the stimulated cells. In addition, all the samples reduced the levels of IL-4 [102]. This study demonstrated that changes in the growing place of plants and, as a consequence, differences in the composition of essential oils, may cause profound changes in their pharmacological activities. Therefore, the location of growing plants has a significant meaning for the development of herbal preparations. However, this fact itself induces great obstacles to this development.

The hydroalcoholic extract of *M. piperita* modulated macrophage-mediated inflammatory responses by activating key antioxidant enzymes and decreasing the levels of superoxide (O_2_^−^) and H_2_O_2_. It is worth noting that the survival of peritoneal macrophage stimulated with lipopolysaccharide was observed with lower concentrations of the extract, while higher concentrations of the extract decreased the viability in the absence of lipopolysaccharides [85]. These biological activities of the hydroalcoholic leaf extract of *M. piperita* may be beneficial if the excessive activation of macrophages leads to tissue damage during infectious disease.

Another study indicated that herbal medicine containing 30–55% of menthol and 14–32% of menthone demonstrated immunomodulatory and antiparasitic activities in the experimental murine model of schistosomiasis by means of reducing the plasma level of IL-4 and IL-10, and diminishing the blood eosinophils after sixty days of treatment that ultimately contributed to the decrease in physiopathological activities. The reduction of the levels of IL-4 and IL-10 was more significant compared to the groups treated with praziquantel. Among the antiparasitic activities was a decrease in the number of eggs in the liver and mesenteric tissue, and reduced liver granuloma formation. Soluble egg antigen stimulated the production of IL-4, IL-5, and IL-13, and led to the upregulation of eosinophils. The coauthors explained the reduced levels of IL-4 and IL-10 after 60 days of the administration of the herbal preparation as a result of the reduced deposition of eggs in the liver and mesenteric tissue [93].

The generalized information about the results of anti-inflammatory activity of essential oil and extracts of *M. piperita* provided by different authors is presented in Table 3.

We suggest that essential oil and extracts of *M. piperita* are promising herbal preparations due to their anti-inflammatory and wound-healing properties for use in skin care products, care products for oral cavities, and even in preparations for systemic administration in combination with synthetic anti-inflammatory or antischistomatosis drug substances in tablets and capsules.

## 7. Analgesic Activity

Analgesic activity is related to anti-inflammatory activity. The analgesic activity of the essential oil of *M. piperita* was assessed using the writhing test in mice. The essential oil showed significant and dose-dependent antinociceptive effects of reducing the number of writhes at all of the three tested doses (2, 20, and 200 μL/kg), with a similar result to phloroglucinol (80 mg/kg), which was the reference drug. An injection of acetic acid gave 77.8 ± 5.4 writhes in the vehicle control group. The administration of phloroglucinol and essential oil at doses of 2, 20, and 200 μL/kg presented 28.53%, 22.87%, 28.53% and 32.13% of antinociceptive activity, respectively, when compared with the vehicle group [60]. A similar activity was observed with *Mentha spicata* [101].

## 8. Antimicrobial Activity

*M. piperita* extracts and essential oil were also reported to exhibit antimicrobial and antiviral activity [12,13,25,70,103,104]. This activity can help to prevent or treat infections at least on the level of complementary therapy. Antimicrobial properties make *M. piperita* extracts and essential oil promising candidates for developing new antimicrobial drug products.

### 8.1. Antiviral Activity

Considering a limited number of antiviral drug substances and the resistance of viruses to them [107], herbal medicinal products can be used for the treatment of viral respiratory infections and other viral infections.

The ethanolic extract of *M. piperita* with relatively high TPC and TFC showed antiviral activity against RSV in Hep-2 cells with an IC_50_ value of 10.4 µg/mL and a selectivity index value of 21.83. The IC_50_ value of the extract was lower compared to that of ribavirin. However, the value of the selectivity index was statistically higher than ribavirin, indicating that *M. piperita* could be regarded as a plant resource with antiviral properties. This ethanolic extract also reduced the synthesis of some inflammatory mediators (NO, TNF-a, interleukin IL-6, and PGE2) in lipopolysaccharide-stimulated RAW 264.7 cells. Phenolics of *M. piperita* may be connected to the antiviral activity of its extracts against RSV [12].

Zeljkovic et al. revealed that the essential oil of *M. piperita* is not active against SARS-CoV-2 [104]. In this study the antiviral effective concentrations (EC50) of essential oils and certain active substances were determined, which provided 50% inhibition of the SARS-CoV-2 induced destruction of infected cells. It was established that carvone and carvacrol had moderate antiviral activity, with EC_50_ concentrations of 80.23 ± 6.07 µM and 86.55 ± 12.73 µM, respectively, while the other monoterpenes were less active (EC_50_ ≤ 100.00 µM). For instance, 1,8-cineole, menthofuran and pulegone had an IC_50_ value of 128.93 ± 14.98 µM, 154.37 ± 16.33 µM, and 195.70 ± 17.12 µM, respectively. In addition, menthol and menthone were inactive against SARS-CoV-2. The structure and activity relationship of the related monoterpenes showed that the presence of keto and hydroxyl groups was associated with the activity of carvone and carvacrol, respectively. As a result, the essential oil of *M. villosa* rich in carvone showed the highest antiviral activity (EC_50_ = 127.0 ± 4.63 ppm), while other essential oils showed mild antimicrobial activity (140 ppm ≤ EC_50_ ≤ 200 ppm) and weak activity (200 ppm ≤ EC_50_). Among the essential oils with weak antiviral activity against SARS-CoV-2 were *M. piperita ꞌPerpetaꞌ, M. piperita ꞌCitrataꞌ*, *M. piperita ꞌBergamotꞌ*, *Thymus vulgaris*, *Satureja montana,* etc. Moreover, carvacrol, menthofuran, thymol, and essential oils with a high content of these monoterpens increased the cytotoxicity of *Vero* cells [104].

A virucidal activity of herbal preparations of some species of the *Lamiaceae* family was revealed for herpes simplex virus (HSV) type 1 (HSV-1) and type 2 (HSV-2), and human immunodeficiency virus type 1 (HIV-1) [13,107]. The essential oil completely inhibited the growth of HSV-1 in vitro at a concentration of 1% [107]. This can indicate its potential therapeutic effect for the treatment of oral herpes.

The aqueous extracts from *Melissa officinalis*, *M. piperita*, and *Salvia officinalis* significantly and rapidly reduced the infectivity of HIV-1 virions at non-cytotoxic concentrations in primary cell models of HIV-1 infection in vitro and ex vivo (the human T-lymphoblastoid cell line Sup-T1, C8166 T-cells, and monocyte-derived macrophages). One of the activity mechanisms was an enhancement in the virion’s density before its surface engagement under the influence of the extracts [13].

### 8.2. Antibacterial and Antifungal Activities

The results of numerous studies showed that the essential oil and extracts of *M. piperita* can inhibit the growth of some bacteria and fungi, indicating its potential antimicrobial activity [1,15,20,25,32,50,61,62,70,78].

The antimicrobial activities of *M. piperita* essential oil against microorganisms were tested using the zone diameter. The results obtained from the disc diffusion method indicated that inhibition zones were 26.3 mm, 15.3 mm, 16.3 mm, 15.0 mm, and 8.3 mm, respectively, for *Candida albicans* ATCC 10231, *Escherichia coli* ATCC 25922, *Streptococcus pyogenes* ATCC 19615, *Staphylococcus aureus* ATCC 25923, and *Pseudomonas aeruginosa* ATCC 27853 when 20 µL of the essential oil was applied. The essential oil of *M. piperita* was lethal to both *Staphylococcus aureus* and *Escherichia coli* [61]. In one more study, the largest mean zone of inhibition was also for *Staphylococcus aureus* (31.1 ± 0.2 mm) [62].

Zaidi and Dahiya investigated the antibacterial activity of *Mentha* species essential oils on eleven bacterial and four fungal clinical isolates using the agar well diffusion method and the TLC-bioautographic method. In this study, the essential oils from *Mentha* species possessed antimicrobial activity against several clinical strains. Both essential oils showed the maximum activity against three isolates of *Staphylococcus aureus*, producing zones of inhibition from 11.4 ± 0.1 to 21.0 ± 0.1 mm for *M. spicata* and from 12.0 ± 0.1 to 19.2 ± 0.1 mm for *M. piperita*. In addition, the peppermint oil also had antifungal activity against *Candida albicans* (11.7 mm) [15]. The essential oil of *M. piperita* had the highest TPC (12.6 ± 0.9 μg GAE), followed by *M. spicata* (9.4 ± 0.6 μg GAE). It is worth noting that the Gram-positive bacteria were slightly more susceptible to the essential oils compared to the Gram-negative bacteria such as *Escherichia coli*, *Pseudomonas aeruginosa*, and *Salmonella* spp [15]. This could be explained by the specific characteristics of the cell envelope of the Gram-negative bacteria [46].

In another study, the minimum inhibitory concentration (MIC) of the essential oil of *M. piperita* was very low for *Staphylococcus aureus* MTCC 902, *Escherichia coli* MTCC 443, and *Candida albicans* ATCC 90028, namely, 0.32, 5.12 and 0.08 mg/mL, respectively [25]. The MIC of the essential oil of *M. piperita* against twenty eight clinical *Staphylococcus aureus* isolates was in the range of 64 to 256 mg/mL and the MIC values of this essential oil against *Staphylococcus aureus* strains ATCC 29213, MRSA 2985, and MRSA 3701 was 128 mg/mL [70].

Singh et al. found in their study that the essential oil of *M. piperita* inhibited the growth of microorganisms and these results were comparable with those of antibiotic gentamycin. The MIC for the bacterial species ranged from 0.4% to 0.7% *v*/*v*. Moreover, this study also demonstrated that the Gram-positive bacteria were more susceptible to the essential oils than the Gram-negative bacteria such as *Escherichia coli* and *Klebsiella pneumonia.* The inhibition zones of 10 µL of the *M. piperita* oil were 17.2 ± 0.9, 13.1 ± 0.7, 5.1 ± 0.4, and 12.4 ± 0.7 mm for *Staphylococcus aureus* ATCC2 5923, *Streptococcus pyogenes* ATCC 19615, *Escherichia coli* ATCC 25922, and *K. pneumonia* ATCC 13883, respectively, compared to gentamycin, with inhibition zones of 14.7 ± 0.4, 18.9 ± 0.7, 19.7 ± 0.3, and 21.2 ± 0.6 mm, respectively. Finally, it was revealed that the antibacterial activity significantly depended on the volume of the essential oil (10, 1, and 0.1 µL) [20].

The MICs of the essential oils of *M. piperita* and *M. arvensis* against *Fusarium moniliforme*, *Aspergillus niger*, and *Aspergillus fumigates* were determined by Nilo et al. using the broth microdilution method. The MIC of the essential oils was in the range of 1.25 to 2.50 μmg/mL [1]. Moreover, Socović et al. revealed that menthol had fungistatic and fungicidal activities, with a MIC of 0.25–1.5 μL/mL in ethanol and 0.05–1.0 μL/mL in Tween, while the minimum fungicidal concentrations for *M. piperita* essential oil were 1.5–3.0 μL/mL and 1.0–2.5 μL/mL [32].

The most notable antibacterial activity exerted by the essential oil of *M. piperita* was against the Gram-positive bacteria *Streptococcus pyogenes* ATCC 19615™. The inhibition zone, MIC, and minimum bactericidal concentration (MBC) were 33.33 mm, 1.25 µg/mL, and 10 µg/mL, respectively. The weakest activity was against the Gram-negative bacteria *P. aeruginosa* ATCC 27853™ (11.33 mm at 10 µg/mL). The antifungal activity was significant against *Candida albicans* ATCC 90029™ and *Candida parapsilosis* ATCC 22019™. The inhibition zones, MIC, and minimum fungicidal concentration (MFC) were 32.33 mm, 1.25 µg/mL, and 10 µg/mL, and 31.33 mm, 1.25 µg/mL and 10 µg/mL, respectively [50].

The growth of *Escherichia coli* and *Staphylococcus aureus* after treatment with 5000 μg/mL of nanogel containing *M. piperita* essential oil was reduced by 100 and 65%, respectively [99]. The MIC of the peppermint essential oil increased in the following order: *Propionibacterium acnes* ATCC 6919 (0.03% *v*/*v*) > *Staphylococcus aureus* ATCC 6538 (0.08% *v*/*v*) = *C. albicans* > *Escherichia coli* ATCC 2592 (0.15% *v*/*v*) > *Pitrosporum ovale* ATCC 12078 (0.22% *v*/*v*) > *Pseudomonas aeruginosa* ATCC 9027 (0.92% *v*/*v*). In this study, the Gram-positive *Staphylococcus aureus* and *Propionibacterium acnes* were also more sensitive to the essential oils than the Gram-negative *Escherichia coli* and *Pseudomonas aeruginosa* [80].

In another study, the essential oil of *M. piperita* had very high activity against *Pseudomonas aeruginosa*, with an MIC and MBC of 6.25 µL/mL (0.625% *v*/*v*), and against *Escherichia coli*, with an MIC and MBC of 50 µL/mL, while the *Candida* strains showed an MIC and MFC in the range of 12.50 µL/mL (1.25% *v*/*v*) to 100 µL/mL (10% *v*/*v*). In accordance with the antibiotic resistance of the clinical isolates, the Gram-negative bacteria *Pseudomonas aeruginosa* and *Escherichia coli* showed multi-resistance to all of the tested antibiotics, which points to the high resistance of clinical isolates to conventional antibiotics [108].

Thus, the essential oil of *M. piperita* can be a good source of natural antimicrobial substances for the development of dosage forms for topical applications for complementary treatment of patients with skin conditions induced by the Gram-positive bacteria (*Staphylococcus aureus, Streptococcus pyogenes)* and fungi, including new anti-acne herbal medicinal products. Considering the susceptibility of *Pseudomonas aeruginosa* to the peppermint essential oil in some studies, there is a possible alternative administration of this essential oil in pessaries for the prevention of infections caused by these vaginal bacteria.

The essential oil of *M. piperita* has an application in dentistry, especially for the prevention of a mineral alteration of the dental tissue. There are a lot of causes that may be responsible for this disease. One of them is a microbial biofilm playing a critical role in the progression of dental decay. The second cause of dental caries is mouth-inhabiting bacteria synthesizing acid by-products as a result of carbohydrate metabolism, which induces dental structure demineralization. The biofilm, as a complicated matrix, consists of proteins and exopolysaccharides which form a physiological barrier for bacterial cells and protect them against active antimicrobial substances. *Streptococcus mutans* is the most well-known biofilm-forming bacterium inhabiting the mouth cavity, and it is associated with dental plaque. The expression of glucosyltransferase genes is considered as a promoting factor of biofilm synthesis as they produce glucan exopolysaccharides which have a critical role in bacterial adhesion and binding to dental surfaces. Therefore, the reduction of biofilm formation by bacteria is considered as a prevention method against bacterial plaque on the dental surface and could decrease acid production [7].

Ashrafi et al. demonstrated that the nanoformulation of peppermint oil loaded in a chitosan nanogel had a great inhibitory action against some glycosyltransferase genes (gtfB, C and D) as important enzymes involved in extracellular polymers. Hydrogels based on chitosan form nanostructural networks that can entrap active substances. In addition, the *M. piperita* essential oil loaded into chitosan nanogel could be regarded a potential antibiofilm agent in toothpaste or mouth washing formulations for the prevention of caries [7]. Notable antimicrobial activity was revealed for *Streptococcus mutans* in another study. The inhibition zone, MIC, and MBC were 21.7–31.7 mm depending on the concentration of the essential oil used (2.5–10 µg/mL), 0.625–1.25 µg/mL, and 0.625–1.25 µg/mL, respectively [50].

## 9. Anticancer Activity

Searching for alternative and complementary medical products from plants is a topical issue for the treatment of cancer. Therefore, there are many studies directed at the study of different plant resources in the search for anticancer active substances or active substances for cancer prevention [19,50,78,81,108,109,110,111,112,113,114].

The results of numerous studies showed that the essential oils and different extracts of several species of *Lamiaceae* can inhibit the growth of cancer cells, indicating the potential effects for the complementary treatment of cancer [19,78,110]. Among the advantages of essential oils in the fight against cancer is their lipophilic character, which facilitates their crossing through cell membranes and reaching the inner side of the cell [50,51,109,110].

Jain et al. tested the different extracts of *M. piperita* on six human cancer cell lines and two normal cell lines: cervical adenocarcinoma (HeLa), breast adenocarcinoma (MCF-7), Jurkat (T-cell lymphoblast), urinary bladder carcinoma (T24), colon adenocarcinoma (HT-29), pancreatic adenocarcinoma (MIAPaCa-2), and the two normal cell lines (lung fibroblast (IMR-90) and kidney epithelial (HEK-293)). The chloroform and ethyl acetate extracts had significant dose- and time-dependent anticarcinogenic activity, inducing G1 cell cycle arrest and mitochondrial-mediated apoptosis, perturbation of oxidative balance, upregulation of pro-apoptotic Bax gene, increased expression of p53 and p21 in the treated cells, and acquisition of the senescence phenotype. In addition, the ethyl acetate extract exerted the largest activity in the generation of ROSs in MIAPaCa-2 cells, while the chloroform extract demonstrated the strongest effect in MCF-7 cells. These two extracts significantly inhibited the antioxidant enzyme glutathione reductase activity in MIAPaCa-2 and T24 cells compared to the controls and elevated the concentrations of the pro-inflammatory cytokines (IL-6, IL-12p70, IL-1b, TNF, IFN-γ, and IL-8) in the culture supernatant of all the cell lines [109].

Another study revealed the prominent cytotoxic effects of the essential oils of four *Mentha* species (*M*. *arvensis, M*. *longifolia, M. piperita,* and *M. spicata*) on the growth of two cancer cell lines (human breast cancer cell line (MCF-7) and hormone-dependent prostate carcinoma LNCaP). The results showed the potential toxic effects of the essential oils as their IC_50_ were in the range of 10–100 µg/mL in the MTT test [112]. The peppermint oil, at concentrations of 0.02, 0.0075, and 0.00125 µL/mL, reduced the number of human cervical carcinoma Hela cells (ATCC number CCL-2) by 98.48%, 97.01%, and 92.64%, respectively [113]. The oral administration of an aqueous extract of *M. piperita* significantly reduced the number of lung tumors from an incidence of 67.92% in the Swiss albino mice given only benzo[a]pyrene to 26.31% [114]. In addition, the essential oil of *M. piperita* of Chinese origin was found to be active against SPC-A-1, K562, and SGC-7901 cancer cell lines [19]. According to Sun et al., the cytotoxic activity of peppermint oil can be induced through the synergic effects of different terpenes, or some other active compounds could be responsible for the cytotoxic activity [19].

However, some studies show a lack of anticancer activity for preparations based on *M. piperita*. The essential oil exhibited strong antimicrobial activity and moderate antioxidant activity, but a low cytotoxic effect in HCT 116 cells (colorectal cancer), while the cytotoxic activity on non-cancerous cell line HaCaT was not significant [50]. In one more study, the aqueous extract of *M. piperita* did not have a clear effect on the apoptosis of human leukemia cells [111]. The peppermint oil was inactive against BEL-7402 cell [19].

Summing up, preparations of *M. piperita* possess anticancer properties of different extents and could be regarded as the basis for further laboratory research concerning the potential use of *M. piperita* preparations or active substances for anticancer treatments.

## 10. Cardiovascular Diseases

*M. piperita* preparations showed hypotensive, vasorelaxant, and antiplatelet activities [26,29,36,113,115,116,117]. The inflammatory activity of *M*. *piperita* may be responsible for diminishing risks of CVDs because patients with these diseases have high inflammation [29,36]. Twenty five students between the age of 18 and 45 years old were tested after an oral administration of *M. piperita* juice two times per day for 30 days. Among the results of this trial were a decrease in glycemia (41.5% of participants), total cholesterol levels (66.9%), triacylglycerides (58.5%), low-density lipoproteins indices (52.3%), glutamic-oxaloacetic transaminase levels (70%), glutamic-pyruvic transaminase levels (74.5%), and urea levels (69.3%). It was found that 52% of the participants had an increase in high-density lipoprotein cholesterol indices, while 52.5% of the students showed a reduced blood pressure, 43.8% showed a weight loss, and 48.7% showed a reduced body mass index [26].

The aqueous extract of *M. piperita* after the oral administration showed significant good effects against fructose-induced hyperlipidemia. There was a decrease in the elevated levels of glucose, cholesterol, triglycerides, very-low-density lipoprotein, low-density lipoprotein, and atherogenic index. Simultaneously, there was an increase in the high-density lipoprotein cholesterol levels and HDL-ratio without affecting serum insulin levels in fructose-fed rats. A decrease in the glutathione level was observed in liver of fructose-fed animals and was considered an indicator of increased oxidative stress. However, the content of glutathione in the liver of fructose-fed rats was increased after an oral administration of the extract [115]. Similar results were obtained in the studies with the rats fed with the essential oil of *M. piperita.* There was a significant decrease in the levels of uric acid, and there was a significant increase in the levels of total cholesterol, triglycerides, and high-density lipoprotein cholesterol [113].

Hypotensive activity due to the reduction in the arterial smooth muscle tonicity after the oral administration of peppermint essential oil by people can explain a decrease in the heart rate and systolic blood pressure [116]. Rosmarinic acid at a dosage of 30 mg/kg improved the cardioprotective effects against acute myocardial infarction and arrhythmia from two consecutive subcutaneous injections of isoproterenol at a dose of 100 mg/kg. This could be explained by the ability of rosmarinic acid to enhance the expression of plasma antioxidant enzymes and genes involved in Ca^2+^ homeostasis and/or its antiadrenergic effects [117].

Summing up, preparations of *M. piperita* have cardiprotective effects due to their antioxidant activities [115], increased high-density lipoprotein cholesterol levels [113,115], and a reduction in arterial smooth muscle tonicity [116], etc. Therefore, preparations of *M. piperita* can be components of medicinal products with cardioprotective activity for the decrease in the blood pressure or prevention of acute myocardial infarction.

## 11. Other Activities (Gastrointestinal Effects, Protective Activity, Larvicidal, and Repellent Activities) and Other Applications

*M. piperita* preparations were reported to have neuroprotective, hepatorenalprotective, and gastrointestinal effects such as anti-spasmodic and anti-ulcer activity, and anti-spasmodic activity of smooth muscles [81,94,118,119,120,121,122,123,124,125].

The essential oil of *M. piperita* exhibited an anti-spasmodic activity on rat trachea involving nitric oxide synthase activation and producing nitric oxide as a major mediator in the neural relaxation of smooth muscles that can be used in the development of herbal medicinal products for the treatment of respiratory diseases [122]. The *M. piperita* aqueous extract reduced the smooth muscle contractions of rat ileum, indicating its anti-spasmodic activity [123].

Another study investigated the effects of *M. piperita* extract on gastric ulcer induced using indomethacin in rats. The results showed that the extract reduced the ulcer index and increased the mucus content in the stomach, indicating its anti-ulcer activity [124]. The pentaherbal medicinal product DCD-684 containing decoctions of five medicinal plants (*Carum carvi* L., *Foeniculum vulgare* Mill, *M. arvensis* L., *M. piperita* L., and *Zingiber officinale* Roscoe) had a significant in vitro muscle relaxant effect on the isolated rabbit jejunum via both non-receptor (calcium channel) and multiple receptor-mediated contractions. Among the individual components, only *M. piperita* significantly inhibited KCl-induced contractions. This study supported the efficacy and therapeutic administration of this product in the management of various gastrointestinal disorders, including infantile colic pain [125].

It seems that extracts of *M. piperita* could be components of combined anti-inflammatory medicinal products, in which these extracts would have the anti-inflammatory and gastroprotective functions, and of combined spasmolytic medicinal products.

*M. piperita* preparations were reported to have neuroprotective activity, which can help to prevent or treat neurodegenerative disorders, such as Alzheimer’s disease and Parkinson’s disease. The nootropic and anti-amnesic activity of peppermint oil was revealed using a rat model of scopolamine-induced amnesia-like Alzheimer’s disease [119]. The improved cognitive function was observed in people after peppermint essential oil inhalation [75]. The essential oil of *M. piperita* inhibited cholinergic activity, regulated calcium levels, and was bound to GABAA/nicotinic receptors in vitro. This essential oil improved the cognitive performance of healthy adults during difficult tasks and reduced the mental fatigue typically associated with prolonged cognitive activity [120].

The essential oil showed an LC_50_ value of 414.6 g/mL against *Artemia salina* and is considered toxic against *A.* aegypti [121]. A nanogel on the base of the essential oil of *M. piperita* had a complete protection time of 120 ± 8 min against *Anopheles stephensi*, the main malaria mosquito vector, while N,N-diethyl-3 methylbenzamide had a protection time of 140 ± 8 min. N,N-diethyl-3 methylbenzamide is one of the best known and most successful synthetic chemical repellents [103].

It is worth noting that extracts of *M. piperita*, as safe mixtures of active substances with a reducing ability, are used in nanotechnology, for instance, for the synthesis of silver nanoparticles. Synthesized nanoparticles of metals using herbal extracts are generally considered safer than their chemically synthesized counterparts [126,127].

## 12. Conclusions

This paper provides a critical descriptive review of the chemical composition of the essential oil and extracts and the pharmacological activities of *M. piperita* from the point of view of the development of new products. The main active component of *M. piperita* is essential oil (0.5–4%), in which the principal constituent is usually menthol, in the form of (−)-menthol, with smaller amounts of its stereoisomers (+)-neomenthol and (+)-iso-menthol. Menthofuran (1.0–8.0%), as a minor component, is considered a specific marker of peppermint as other species of *Mentha* do not contain it. Reducing levels of menthofuran and pulegone has commercial significance in improving the quality of the essential oil of *M. piperita*. *M. piperita* extracts and essential oil have a wide range of biological activities, such as antioxidant, anti-inflammatory, antimicrobial, cardioprotective activities, etc. *M. piperita* can be a platform for the future development of potent and safer herbal medicinal products or combined medicinal products (active pharmaceutical ingredients with herbal preparations). *M. piperita* preparations have antimicrobial activity and could increase innate immunity and decrease active oxygen species levels. Therefore, this plant could be used as the raw material for the manufacture of herbal products with antioxidant, anti-inflammatory, antimicrobial, cardioprotective, gastroprotective, and immunomodulatory activities. The results of this review support the use of peppermint essential oil and extracts in the development of pharmaceutical products for the management of inflammation and pain. However, more studies are needed to fully understand their mechanisms of action and potential applications in the prevention and/or treatment of various diseases. This plant also could be used as a functional ingredient in the food processing industry.

## Figures and Tables

**Figure 1 molecules-28-07444-f001:**
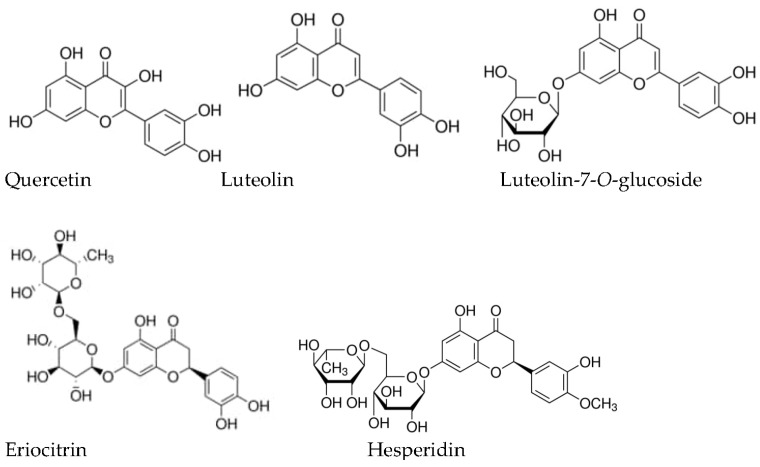
The chemical structure of main flavonoids of *M. piperira*.

**Table 1 molecules-28-07444-t001:** Chromatographic profiles of essential oil of *M. piperita* specified by international standards.

Component	Specified Components [%]			
	[81]		Ph. Eur. [53]	WHO [83]
	Non-US Origin	US Origin		
menthol	32.0–49.0	36.0–46.0	30.0–55.0	30.0–55.0
menthone	13.0–28.0	15.0–25.0	14.0–32.0	14.0–32.0
isomenthone	2.0–8.0	2.0–4.5	1.5–10.0	2.0–10
menthyl acetate	2.0–8.0	3.0–6.5	2.8–10.0	3.0–5.0
eucalyptol (1,8-cineole)	3.0–8.0	4.0–6.0	3.5–8.0	6.0–14.0
menthofuran	1.0–8.0	1.5–6.0	1.0–8.0	1.0–9.0
neomenthol	2.0–6.0	2.5–4.5		
limonene	1.0–3.0	1.0–2.5	1.0–3.5	1.0–5.0
*trans*-sabinene hydrate	0.5–2.0	0.5–2.3		
pulegone	0.5–3.0	0.5–2.5	0–3.0	0–4.0
β-caryophyllene	1.0–3.5	1.0–2.5		
3-octanol	0.1–0.5	0.1–0.4		
carvone			0–1.0	0–1.0
1,8-cineole/limonene ratio				>2.0

**Table 2 molecules-28-07444-t002:** Information about the results of antioxidant activity of essential oil and extracts of *M. piperita*.

Test	Tested Material, Origin	Values	Values for a Reference Substance	Reference
DPPH	Essential oil, Egypt	IC_50_ = 59.2 µg/mL	IC_50_ for TBHQ 29.8 µg/mL	[8]
DPPH	Essential oil Aqueous extract, Brazil	IC_50_ = 13.6 mg/mL IC_50_ = 12.2 mg/mL	No data	[1]
Pulse voltammetry	Essential oil, Brazil	Rate constants 155.9 mL/g,122.4 mL/g		[9]
FRAP	Methanol/chloroform (3:1) extract, Turkey	317.60 ± 49.32 558.33 ± 13.52 μmol trolox equivalents per one gram of dry weight	Trolox	[98]
TEAC		771.58 ± 3.22 and 800 ± 10 ± 1.10 μmol trolox equivalents per one gram of dry weight	Trolox	
DPPH	Essential oil, Midwest region of the USA	IC_50_ 70.29 mg/mL	No data	[16]
TEAC		IC_50_ 29.51 mg/mL		
Reducing power assay		IC_50_ 22.7 mg/mL		
Lipid peroxidation assay in pig liver homogenate		Reducing lipid peroxidation in a dose-dependent manner at 1000 µg/mL; at a concentration of 2000 µg/mL there was no reducing lipid peroxidation	Malondialdehyde	
Cellular antioxidant activity in jejunal epithelial cell line IPEC-J2 with DCF		Maximal inhibitory effect at a concentration of 5 µg/mL	Trolox	
Intracellular antioxidant activity for glutathion (GSH)		Essential oil did not enhance GSH production.		
In vivo antioxidant analysis with nematode model		The survival rate was increased at the concentrations 10, 25, 50, and 100 µg/mL; at a concentration of 200 µg/mL the survival was on the level of the control	Trolox	
Test with peroxidase	Essential oil Chloroform extract Ethanolic extract Aqueous extract, Libya	89.4% 91.2% 76.2% 69.8%		[20]
DPPH	Essential oil Chloroform extract Ethanolic extract Aqueous extract	92.6% (IC_50_ = 15.2 µg/mL) 91.8% 74.8% 70.3%	IC_50_ for BHT 6.1 µg/mL	
Reducing power assay, absorbance at 700 nm	Essential oil Chloroform extract Ethanolic extract Aqueous extract	0.9 ± 0.3 0.8 ± 0.3 0.7 ± 0.1 0.4 ± 0.3		
DPPH	Ethanolic extract, Iran	IC_50_ 13.32 µg/mL (12.12–14.64 µg/mL)	rutin, IC_50_ 6.90 (6.61–7.20) μg/mL,	[99]
ABTS		IC_50_ 153.80 µg/mL (139.90–169.00 µg/mL)	rutin, IC_50_ 79.59 (69.73–90.86) μg/mL	
DPPH	Essential oil, China	Peppermint concentrations were 200, 400, 600, 800, and 1000 mg/mL. The respective scavenging capacities ranged from 36.81% to 79.85%	BHT, from 82.36% to 93.85%	[19]

**Table 3 molecules-28-07444-t003:** Information about the anti-inflammatory activity of essential oil and extracts of *M. piperita*.

Table	Type of a Preparation, Doses Tested	Pharmacological Mechanisms of Action	Values	Reference
Carrageenan-induced paw edema test	Essential oil, oral administration 30 min before introduction of carrageenan, 2, 20, and 200 μL/kg, vehicle (isosaline NaCl 0.9%), and sodium diclofenac (50 mg/kg, orally) as the reference drug	Reducing the paw edema induced using carrageenan injection in mice	The levels of edema inhibition,12.27 ± 3.94% and 9.29 ± 3.94%, at 200 and 20 μL/kg, respectively, were comparable to the level observed using sodium diclofenac (11.43 ± 6.07%)	[60]
Circular excision wound model in rats followed by histological examination	The cream prepared from the essential oil (0.5% *w*/*w*)	Decreasing in unhealed wound area	Significant decrease in unhealed wound area between the 6th (1.67 ± 0.14 mm^2^) and 9th (0.49 ± 0.22 mm^2^) day of treatment in comparison with the vehicle and madecassol on 6th day (2.32 ± 0.77 mm^2^ and 2.23 ± 0.35 mm^2^, respectively)	
Murine macrophage cell line RAW 264.7 stimulated with LPS	The ethanolic extract of *M. piperita*, 25, 50, and 100 µg/mL	Reducing the LPS-induced production of NO, TNF-a, and IL-6 compared with untreated control	Reducing NO secretion in a concentration-dependent manner by 7.06, 18.85, and 41.88%, respectively. Suppression of TNF-a secretion by 20.71, 34.74, and 42.95%. Reducing IL-6 levels by 27.00, 43.71, and 51.85%. Inhibition of PGE2 production.	[12]
PBMCs	Essential oil, 0.01 µL/mL	Reducing IL-4 production depends on cultivar and place of growing.		[102]
5-LOX inhibition assay	Essential oil	5-LOX inhibition	IC_50_ were 0.08 and 0.03 μL/mL depending on the cultivar	[80]
Activating antioxidant enzymes, the survival of peritoneal macrophage stimulated with lipopolysaccharide	The hydroalcoholic leaf extract of *M. piperita* grown in Brazil	Activating superoxide dismutase and glutathione peroxidase. Lowering the levels of H_2_O_2_. The survival was observed at the lower concentrations of the extract (1–30 µg/mL), while higher concentrations of the extract did decrease viability in the absence of lipopolysaccharides.	No data	[85]

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
