# Peer review of "Mentha piperita: Essential Oil and Extracts, Their Biological Activities, and Perspectives on the Development of New Medicinal and Cosmetic Products"

_molecules, 2023, doi:10.3390/molecules28217444_

Round 1
Reviewer 1 Report
Comments and Suggestions for Authors
Some questions regarding the manuscript are presented:
1- I suggest a modification in the title of the manuscript, considering that antioxidant activity also fits into a biological activity.
2- A dynamic reading of the work is necessary, given that some statements seem to be loose throughout the manuscript.
3- Observe the abstract carefully, as they have some writing errors (wrongly written words, plant species are not in italics).
4- In the introduction, several botanical species are mentioned, however, without the proper botanical authorities. Example: Salvia apiana. Corrector would be Salvia apiana Jeps.
5- It is necessary that the authors insert photographs exemplifying the differences of Mentha piperita, M. pulegium, M. citrate and M. spicata. M. pulegium.
6- I suggest inserting a paragraph addressing the commercial issue of the plant species (how much it generates per year, which commercial forms are found...).
7- Standardize throughout the text the nomenclature and popular name of the species (peppermint oil, essential oil of M. piperita, menthol mint). This creates confusion for the reader.
8- I suggest that the authors insert a table in the antioxidant topic, listing the methods employed, the values obtained, the major components found (essential oil), extraction method.
9- In the topic of anti-inflammatory activity, it is also necessary to insert a table indicating the methods used for evaluation, the doses tested, the pharmacological mechanisms of action found, the major components found (essential oil).
10- Is the species in question not used as an analgesic? I suggest the insertion of one more topic below the anti-inflammatory activity.
11- The topic of antimicrobial activity was not addressed correctly. Which species of pathogens tested (human, animal, or plant). Which methods are used for the analyses. What values were found?
12- The topic Dentistry should be addressed within the antibacterial activity, considering that its description is associated with pathogens of the oral cavity.
13- I suggest that the authors bring articles related to the development and technological applications using biomolecules from the plant in question: (https://doi.org/10.1016/j.lfs.2018.08.046, https://doi.org/10.1016/j.inoche.2022.109581, https://doi.org/10.1016/j.foodchem.2012.10.106 , https://doi.org/10.1016/j.foodchem.2015.05.108 , https://doi.org/10.1016/j.foodcont.2011.04.002)
It is necessary that these modifications be made so that the manuscript can be accepted for publication.
Comments on the Quality of English LanguageSome questions regarding the manuscript are presented:
1- I suggest a modification in the title of the manuscript, considering that antioxidant activity also fits into a biological activity.
2- A dynamic reading of the work is necessary, given that some statements seem to be loose throughout the manuscript.
3- Observe the abstract carefully, as they have some writing errors (wrongly written words, plant species are not in italics).
4- In the introduction, several botanical species are mentioned, however, without the proper botanical authorities. Example: Salvia apiana. Corrector would be Salvia apiana Jeps.
5- It is necessary that the authors insert photographs exemplifying the differences of Mentha piperita, M. pulegium, M. citrate and M. spicata. M. pulegium.
6- I suggest inserting a paragraph addressing the commercial issue of the plant species (how much it generates per year, which commercial forms are found...).
7- Standardize throughout the text the nomenclature and popular name of the species (peppermint oil, essential oil of M. piperita, menthol mint). This creates confusion for the reader.
8- I suggest that the authors insert a table in the antioxidant topic, listing the methods employed, the values obtained, the major components found (essential oil), extraction method.
9- In the topic of anti-inflammatory activity, it is also necessary to insert a table indicating the methods used for evaluation, the doses tested, the pharmacological mechanisms of action found, the major components found (essential oil).
10- Is the species in question not used as an analgesic? I suggest the insertion of one more topic below the anti-inflammatory activity.
11- The topic of antimicrobial activity was not addressed correctly. Which species of pathogens tested (human, animal, or plant). Which methods are used for the analyses. What values were found?
12- The topic Dentistry should be addressed within the antibacterial activity, considering that its description is associated with pathogens of the oral cavity.
13- I suggest that the authors bring articles related to the development and technological applications using biomolecules from the plant in question: (https://doi.org/10.1016/j.lfs.2018.08.046, https://doi.org/10.1016/j.inoche.2022.109581, https://doi.org/10.1016/j.foodchem.2012.10.106 , https://doi.org/10.1016/j.foodchem.2015.05.108 , https://doi.org/10.1016/j.foodcont.2011.04.002)
It is necessary that these modifications be made so that the manuscript can be accepted for publication.
Author Response
Dear reviewer, thank you for your hard work with our manuscript. Your valuable remarks and recommendations, we hope, have improved our manuscript significantly.
1. I suggest a modification in the title of the manuscript, considering that antioxidant activity also fits into a biological activity.
We totally agree with this recommendation. We deleted antioxidant in the title.
- A dynamic reading of the work is necessary, given that some statements seem to be loose throughout the manuscript.
We totally agree with this recommendation. We hope we improved our manuscript by working with the remarks of all four reviewers. If you insist on further dynamic reading, we will continue.
- Observe the abstract carefully, as they have some writing errors (wrongly written words, plant species are not in italics).
Thank you for this valuable remark. It is done
4.In the introduction, several botanical species are mentioned,however, without the proper botanical authorities. Example: Salvia apiana. Corrector would be Salvia apiana Jeps.
Thank you for this valuable remark. It is done according to The Plant List
http://www.theplantlist.org/tpl1.1/search?q=salvia+officinalis
and https://www.worldfloraonline.org/search?query=Mentha+citrata
Salvia apiana Jeps., Salvia officinalis L., Salvia sclarea L., Thymus vulgaris L., Rosmarinus officinalis L., Prunella vulgaris L., Mentha spicata L., Mentha pulegium L., Mentha × citrata Ehrh., etc.
- It is necessary that the authors insert photographs exemplifying the differences of Mentha piperita, M. pulegium, M. citrate and M.spicata. M. pulegium.
Dear reviewer, we ask you not to consider this remark. Firstly, the paper is too large. Secondly, the purpose of this paper was not botanical characteristics of different species of the genus Mentha. The primary purpose was to collect the information as a base for the development of pharmaceutical products on the base of Mentha piperita. However, if you insist on this issue, we will think about what to do.
- I suggest inserting a paragraph addressing the commercial issue of the plant species (how much it generates per year, which commercial forms are found...).
Dear reviewer, we ask you not to consider this remark. Firstly, the paper is too large. However, if you insist on this issue, we will think about what to do.
7.Standardize throughout the text the nomenclature and popular name of the species (peppermint oil, essential oil of M. piperita, menthol mint). This creates confusion for the reader.
We totally agree. As a corresponding author, I have chosen such nomenclature like this, essential oil of M. piperita,
8.I suggest that the authors insert a table in the antioxidant topic,listing the methods employed, the values obtained, the major components found (essential oil), extraction method.
We totally agree with this remark. Yes, it will better visualize results. We did not provide information about major compounds, because there is no correlation between the composition and activity
9. In the topic of anti-inflammatory activity, it is also necessary to insert a table indicating the methods used for evaluation, the doses tested, the pharmacological mechanisms of action found, the major components found (essential oil).
We agree with this remark. Yes, it will better visualize results as well. We did not provide information about major compounds, because the complex of active substances act, to our mind
10. Is the species in question not used as an analgesic? I suggest the insertion of one more topic below the anti-inflammatory
We agree with this remark.
11. The topic of antimicrobial activity was not addressed correctly.Which species of pathogens tested (human, animal, or plant). Which methods are used for the analyses? What values were found?
We partly agree with this remark. We clarified some information by adding methods and the exact identification of microorganisms like this Candida albicans ATCC10231. These pathogens are human
12. The topic Dentistry should be addressed within the antibacterial activity, considering that its description is associated with pathogens of the oral cavity.
We agree with this recommendations
13- I suggest that the authors bring articles related to the development and technological applications using biomolecules from the plant in question: (https://doi.org/10.1016/j.lfs.2018.08.046, https://doi.org/10.1016/j.inoche.2022.109581, https://doi.org/10.1016/j.foodchem.2012.10.106 , https://doi.org/10.1016/j.foodchem.2015.05.108 , https://doi.org/10.1016/j.foodcont.2011.04.002)
Thank you for this recommendation. We added 2 papers about green nanotechnology
Reviewer 2 Report
Comments and Suggestions for Authors
check language well
authors should write about the cultivation and production of Mentha piperita using some growth regulators or biofertilizers for increasing the yield , and affected in essential oil constituents
write about the effect of biotic and abiotic stress.
Comments on the Quality of English Languagecheck language well
Author Response
Dear reviewer, thank you for your hard work with our manuscript. Your valuable remarks and recommendations, we hope, have improved our manuscript.
Authors should write about the cultivation and production of Mentha piperita using some growth regulators or biofertilizers to increase the yield , and affected in essential oil constituents and write about the effect of biotic and abiotic stress.
We considered the issue of the influence of some growth regulators or biofertilizers on the chemical composition of essential oil.
Dear reviewer, let's not consider the question about biotic and abiotic stress. The paper is so large and dedicated more pharmaceutical and pharmacological aspects. But if you insist on this issue, we will think about what to do.
The language is improved
Reviewer 3 Report
Comments and Suggestions for Authors
Mentha piperita L. is regarded as one of the best potential sources of natural antioxidants for the food and pharmaceutical industries. This manuscript provides a review of the chemical composition and biological activity of the essential oils and the extracts from this plant. But this plant has already reported multiple literature reviews on essential oils and extracts, so the novelty of the manuscript is relatively low. And insufficient correlation between the essential oils and the extracts. Moreover, the logic of the introduction is not clear. All compounds should be numbered in the order. Please provide the stereochemistry of all compounds.
Author Response
Dear reviewer, thank you for your hard work with our manuscript. Your valuable remarks and recommendations, we hope, have improved our manuscript significantly.
Mentha piperita L. is regarded as one of the best potential sources of natural antioxidants for the food and pharmaceuticalindustries. This manuscript provides a review of the chemical composition and biological activity of the essential oils and the extracts from this plant. But this plant has already reported multiple literature reviews on essential oils and extracts, so the novelty of the manuscript is relatively low. And insufficient correlation between the essential oils and the extracts. Moreover, the logic of the introduction is not clear. All compounds should be numbered in the order. Please provide the stereochemistry of allcompounds.
Dear reviewer, we totally agree that there are literature reviews on essential oils and extracts of Mentha piperita. Our primary (main) purpose was to prepare the review from the point of view of the generalization of the information for the development of new products, especially pharmaceutical ones. According to the modern requirement for pharmaceutical development, it is necessary to justify the composition and technology of preparation considering nosology, namely we create this review as a base for the justification of certain compositions.
We cannot try to compare essential oil and extracts. We just described them as the main herbal preparations of Mentha piperita not trying to compare them at all.
We hope that we improved the introduction.
We provide the stereochemistry of all the compounds of essential oil.
All compounds should be numbered in the order – we do not understand your remark
Reviewer 4 Report
Comments and Suggestions for Authors
In this review, Nataliia Hudz and co-authors present a comprehensive assessment of the properties, medical significance, preparation, and extraction techniques of essential oil (EO) and other extracts from the extensively studied plant species Mentha piperita.
This review holds relevance not only for researchers focusing on this plant but also for those in the fields of phytochemistry and phytomedicine.
Nevertheless, I have a general observation regarding this manuscript: In my view, the text is overly lengthy and lacks sufficient focus on the plant under study. Throughout the various sections, numerous sentences and paragraphs are repetitive and/or stray from the main scope. Some instances of this observation are highlighted below.
In conclusion, I strongly recommend that the authors significantly condense their manuscript, retaining only essential and scientifically validated data, and presenting such information in tabular formats when feasible.
Abstract: Biological properties are redundantly listed in the same paragraph: Lines 81 to 88, 99-102, and then 115-122.
Chemical Composition of Essential Oil: The fact that menthol is the principal component of the EO is reiterated in paragraphs at lines 186, 216, 236, and 305.
The comparison with the chemical composition of other species (lines 234-251) is not particularly informative in this context, but rather confusing for the reader. Summarizing this data in a table (as is partly done) would be more effective. In the paragraph at lines 305-334, chemical and biological considerations become intertwined.
Chemical Composition of M. piperita Extracts: Similarly, a table would enhance the value of comparing the various extraction methods and outcomes. In the paragraph spanning lines 368-383, the discussion of antioxidant properties is misplaced.
Anti-oxidant Activity: This section needs to be shortened by omitting general considerations on the link between diseases and ROS (lines 442-454), as well as descriptions of methods for their detection. The amalgamation of general chemical considerations and discussions of various data in the same paragraph is perplexing (see, for instance, lines 506-514). It would be more engaging for readers to focus more on the effects of M. piperita extract.
Anti-inflammatory Activity: As above, I believe this section should be condensed. General considerations on inflammation (lines 655-695) are unnecessary, and presenting essential results in a table format would be more valuable.
Immunomodulatory Effects: The description of "Immunomodulatory effects" (lines 739-794) is bewildering. Data from oils of different origins, herbal extracts, and general immunological or biological considerations are summarized. It's uncertain whether readers can easily access this information.
Anticancer Activity: Is the discussion about the lack of effect on colorectal cancer necessary? Has the effect on other types of cancer been investigated?
Comments on the Quality of English LanguageModerate editing of English language required
Author Response
Dear reviewer, thank you for your hard work with our manuscript. Your valuable remarks and recommendations, we hope, have improved our manuscript. Special thanks for the stylistic remarks because we eliminated a lot of repetitions due to your recommendations.
In this review, Nataliia Hudz and co-authors present a comprehensive assessment of the properties, medical significance, preparation, and extraction techniques of essential oil (EO) and other extracts from the extensively studied plantspecies Mentha piperita.
This review holds relevance not only for researchers focusing on this plant but also for those in the fields of phytochemistry and phytomedicine.
Nevertheless, I have a general observation regarding this manuscript: In my view, the text is overly lengthy and lacks sufficient focus on the plant under study. Throughout the various sections, numerous sentences and paragraphs are repetitiveand/or stray from the main scope. Some instances of this observation are highlighted below.
In conclusion, I strongly recommend that the authors significantly condense their manuscript, retaining only essential and scientifically validated data, and presenting such information in tabular formats when feasible.
Dear reviewer, we totally agree with you, especially concerning the repetition of some phrases and condensing text.
Abstract: Biological properties are redundantly listed in the same paragraph: Lines 81 to 88, 99-102, and then 115-122.
We totally agree with these remarks. We have corrected significantly it.
Chemical Composition of Essential Oil: The fact that menthol is the principal component of the EO is reiterated in paragraphs at lines 186, 216, 236, and 305.
The comparison with the chemical composition of other species (lines 234-251) is not particularly informative in this context, but rather confusing for the reader. Summarizing this data in a table(as is partly done) would be more effective. In the paragraph at lines 305-334, chemical and biological considerations become intertwined.
We agree with these remarks. We have corrected significantly it, eliminating other species. The sense of the remaining paragraph is to show intertwining isomers of menthols in the papers (lines 234-251)
Yes, it is true that chemical and biological considerations become intertwined in lines 305-334 in order to show why it is preferred that essential oil of Mentha piperita contains a high concentration of menthol, moderate levels of menthone, and low amounts of pulegone and menthofuran. We divided these lines into two paragraphs in order to reduce the level of intertwining
Chemical Composition of M. piperita extracts: Similarly, at able would enhance the value of comparing the various extraction methods and outcomes. In the paragraph spanning lines 368-383, the discussion of antioxidant properties is misplaced.
Thank you for these remarks. The part concerning antioxidant properties was transferred into the subchapter Antioxidant activities. Moreover, we generalized antioxidant activities in the form of a table as one more reviewer recommended.
Anti-oxidant Activity: This section needs to be shortened by omitting general considerations on the link between diseases and ROS (lines 442-454), as well as descriptions of methods for their detection. The amalgamation of general chemical considerations and discussions of various data in the same paragraph is perplexing (see, for instance, lines 506-514). It would be more engaging for readers to focus more on the effects of M. piperita extract.
Yes, we agree with this remark
Anti-inflammatory Activity: As above, I believe this section should be condensed. General considerations on inflammation(lines 655-695) are unnecessary, and presenting essential results in a table format would be more valuable.
Yes, we totally agree with this remark
Immunomodulatory Effects: The description of"Immunomodulatory effects" (lines 739-794) is bewildering. Data from oils of different origins, herbal extracts, and general immunological or biological considerations are summarized. It's uncertain whether readers can easily access this information.
We improved this very complicated paragraph
Anticancer Activity: Is the discussion about the lack of effect on colorectal cancer necessary? Has the impact on other types of cancer been investigated?
In this cited paper, there is a lack of anticancer activity. We added some information about other cancer cells.
Dear reviewer, we tried to condense the text mainly for the elimination of repetitions. If you insist on larger condensation, we will consider this issue. Moreover, other reviews recommended to take new aspects.
Language is improved. Thank you one more time for your valuable reccomendations

Round 2
Reviewer 2 Report
Comments and Suggestions for Authors
this article is review so it must include everything related to the topic.
biotic and abiotic stress is very important
Comments on the Quality of English Languageagain check language
Author Response
Dear reviewer, thank you for your recommendations. It is indeed necessary to show the influence of abiotic and biotic factors. They are in the text of the manuscript in blue colour
Reviewer 3 Report
Comments and Suggestions for Authors
This manuscript has been appropriately revised.
Author Response
Dear reviewer, Thank you for your evaluation.
Reviewer 4 Report
Comments and Suggestions for Authors
I thank you for your efforts in considering my previous comments.
Undoubtedly, the inclusion of tables has significantly enhanced the clarity of your objectives and will undoubtedly facilitate future readers' engagement with this review. However, I would like to reiterate my previous observation that certain paragraphs have not been sufficiently condensed. For instance:
-
In the revised version, the section on anti-inflammatory activity (originally spanning lines 655 to 695) remains lengthy (lines 668 to 706). Similarly, this issue persists in the case of anti-cancer activity. Therefore, I reiterate my recommendation to streamline the text, allowing readers to easily discern the essence of your message.
To illustrate, consider simplifying the presentation of inflammation (lines 668-694) by citing two or more general papers, such as:
-
Bryan Oronsky, Scott Caroen, and Tony Reid, ‘What Exactly Is Inflammation (and What Is It Not?)’, International Journal of Molecular Sciences, 23, no. 23 (28 November 2022): 14905.
-
Linlin Chen et al., ‘Inflammatory Responses and Inflammation-Associated Diseases in Organs’, Oncotarget, 9, no. 6 (14 December 2017): 7204–18.
This approach replaces a complex presentation with references that interested readers can explore for comprehensive and accessible information on the phenomenon, allowing you to focus more on the specificity of EO.
Furthermore, consider integrating Arruda et al.'s results into the table and eliminating the textual description of their work. While this is just one example, it can be applied throughout the manuscript.
-
-
The paragraphs introducing both anticancer activity and cardiovascular disease could be succinctly summarized by citing one or two general references.
-
In the same vein, the paragraph discussing anticancer activity need not reiterate all the details from the abstracts of cited papers (e.g., lines 987-1008 and reference 102).
I understand that conveying a clear message with minimal text is a challenging task, but it is crucial to ensure that your work is read and cited by colleagues interested in the expanding field of essential oils.
Author Response
Dear reviewer, thank you very much for your valuable recommendations to improve our manuscript. If you consider one more correction, we will agree to correct this manuscript
We reduced this section and put the reference
- Bryan Oronsky, Scott Caroen, and Tony Reid, ‘What Exactly Is Inflammation (and What Is It Not?)’, International Journal of Molecular Sciences, 23, no. 23 (28 November 2022): 14905.
I thank you for your efforts in considering my previous comments.
Undoubtedly, the inclusion of tables has significantly enhanced the clarity of your objectives and will undoubtedly facilitate future readers' engagement with this review. However, I would like to reiterate my previous observation that certain paragraphs have not been sufficiently condensed. For instance:
-
In the revised version, the section on anti-inflammatory activity (originally spanning lines 655 to 695) remains lengthy (lines 668 to 706). Similarly, this issue persists in the case of anti-cancer activity. Therefore, I reiterate my recommendation to streamline the text, allowing readers to easily discern the essence of your message.
To illustrate, consider simplifying the presentation of inflammation (lines 668-694) by citing two or more general papers, such as:
-
Bryan Oronsky, Scott Caroen, and Tony Reid, ‘What Exactly Is Inflammation (and What Is It Not?)’, International Journal of Molecular Sciences, 23, no. 23 (28 November 2022): 14905.
-
Linlin Chen et al., ‘Inflammatory Responses and Inflammation-Associated Diseases in Organs’, Oncotarget, 9, no. 6 (14 December 2017): 7204–18.2.
-
This approach replaces a complex presentation with references that interested readers can explore for comprehensive and accessible information on the phenomenon, allowing you to focus more on the specificity of EO.
Furthermore, consider integrating Arruda et al.'s results into the table and eliminating the textual description of their work. While this is just one example, it can be applied throughout the manuscript.
-
The paragraphs introducing both anticancer activity and cardiovascular disease could be succinctly summarized by citing one or two general references.
-
In the same vein, the paragraph discussing anticancer activity need not reiterate all the details from the abstracts of cited papers (e.g., lines 987-1008 and reference 102).
-
-
I understand that conveying a clear message with minimal text is a challenging task, but it is crucial to ensure that your work is read and cited by colleagues interested in the expanding field of essential oils.
We tried to correct the whole manuscript, conveying a clear message with minimal text. We hope that we significantly improved our manuscript and even increased the % of plagiarism. Thank you for your valuable recommendations
-
